# Adaptive rewiring shapes structure and stability in a three-guild herbivore-plant-pollinator network

Min Su [1✉], Qi Ma [1] & Cang Hui [2,3,4✉]

Animal species, encompassing both pollinators and herbivores, exhibit a preference for plants based on optimal foraging theory. Understanding the intricacies of these adaptive plant-animal interactions in the context of community assembly poses a main challenge in ecology. This study delves into the impact of adaptive interaction rewiring between species belonging to different guilds on the structure and stability of a 3-guild ecological network, incorporating both mutualistic and antagonistic interactions. Our findings reveal that adaptive rewiring results in sub-networks becoming more nested and compartmentalized. Furthermore, the rewiring of interactions uncovers a positive correlation between a plant's generalism concerning both pollinators and herbivores. Additionally, there is a positive correlation between a plant's degree centrality and its energy budget. Although network stability does not exhibit a clear relationship with non-random structures, it is primarily influenced by the balance of multiple interaction strengths. In summary, our results underscore the significance of adaptive interaction rewiring in shaping the structure of 3-guild networks. They emphasize the importance of considering the balance of multiple interactions for the stability of adaptive networks, providing valuable insights into the complex dynamics of ecological communities.

[1] School of Mathematics, Hefei University of Technology, Hefei 230009, China. [2] Centre for Invasion Biology, Department of Mathematical Sciences, Stellenbosch University, Stellenbosch 7602, South Africa. [3] Mathematical Biosciences Unit, African Institute for Mathematical Sciences, Cape Town 7945, South Africa. [4] International Initiative for Theoretical Ecology, London N1 2EE, UK.  ✉email: sum04@163.com; chui@sun.ac.za

Plant-animal interactions encompass a spectrum, ranging from mutualistic engagements, such as those involving plants and pollinators, to antagonistic encounters, as seen between plants and herbivores[1–4]. Recognizing the joint significance of these interactions is crucial for comprehending the structure, stability, and functioning of ecological communities[2–9]. A growing body of evidence points to the intricate relationship between network stability, measured by parameters like resilience and species persistence, and various factors, including network complexity (size and connectance)[3,10], the balance of multiple interaction types[11,12], and structural features like nestedness and modularity[13,14].

Throughout an animal's life, there is a propensity to strategically shift plant resources, optimizing performance and reflecting an adaptive behavior aimed at enhancing the efficiency of plant resource utilization[14]. This adaptive interaction rewiring, where interactions are rearranged over time due to species altering their interaction partners, has been projected to influence species interactions and impact the stability of both plant-pollinator mutualistic networks[14–16] and plant-herbivore antagonistic networks[17,18]. However, existing research often silos the study of these distinct plant-animal interaction types, potentially obscuring their interconnected feedbacks within real and intricate communities[15,17]. Consequently, a comprehensive understanding of network stability, influenced by the adaptive rewiring of different plant-animal interactions, is still evolving.

The majority of species in natural communities engage in various interspecific interactions, creating interconnected networks[1–3,19–24]. Specifically, flowering plants, as primary producers, engage in reciprocal relationships with pollinators, such as bees, while simultaneously facing challenges from herbivores (e.g., caterpillars) and competitors through antagonistic and competitive interactions[19,24]. This 3-guild network, consisting of pollinators, plants, and herbivores, includes both a mutualistic sub-network and an antagonistic sub-network[1,4,21]. The merging of networks with diverse interaction types establishes crucial pathways for direct and indirect feedback, profoundly influencing the resulting network structure and stability[2–6,21–26].

For example, research by Sauve et al.[3] has demonstrated that the 3-guild network can trigger indirect cascade effects influenced by interconnecting plant species, mitigating the impact of network structure (e.g., nestedness and modularity) on stability. Similarly, Sauve et al.[4] propose that the way plants connect pollinators and herbivores, specifically a positive correlation between a plant's generalism (node degree) in both mutualistic and antagonistic sub-networks, promotes network stability. The composition and balance of strengths among different interaction types in merged networks also play a crucial role in affecting network stability[8,12]. Theoretical studies indicate that there might be an "optimal" way of blending mutualistic and antagonistic interaction types to maximize network stability[8]. Consequently, the merging of pollinator-plant and plant-herbivore interactions within an ecological network could serve as a model framework for understanding the emergence of structure and the complexity-stability relationship. This approach may offer insights distinct from those applicable to networks featuring a single mutualistic or antagonistic interaction[2–4].

Another crucial determinant is the shift in interspecific interactions driven by adaptive foraging, a key in predicting alterations in network structure and stability[15,16,26–30]. This process is often a response to changes in the ecological and environmental context[31–33]. Substantial evidence indicates that adaptive interaction rewiring observably contributes to the persistence of food webs and the resilience of antagonistic networks[17,18,26,27]. In mutualistic networks, interaction switches have been demonstrated to enhance system robustness against species loss and fortify nested architecture, thereby improving network stability[14,27,28]. Nevertheless, recent studies have updated the adaptive rewiring theory, proposing that adaptive mutualistic networks may exhibit less resilience compared to random interaction networks (i.e., those with randomly assigned interactions)[15,16]. Furthermore, adaptive rewiring can alter network stability (measured by resilience) in response to the strength of species interactions. For instance, mutualism strength can either enhance or reduce the stability of an adaptive network of pollinator-plant interactions, depending on competition strength, while mutualism only generates a negative effect on stability in the random interaction network[16].

Considering such adaptive interaction rewiring provides valuable insights into unraveling the structure and stability of ecological networks. Given that the 3-guild network functions as a multiplex network, with plant species serving as interconnecting elements in the pollination and herbivory sub-networks, a key contrast between adaptive and random interaction networks can be elucidated by the distinct roles that a plant species plays in the two sub-networks. Consequently, we anticipate that adaptive interaction rewiring, in conjunction with the balance of multiple species interactions, can give rise to structural properties in 3-guild networks, thereby reshaping the complexity-stability relationship of multiplex networks.

In our endeavor to advance multiplex network theory, this study delves into the structure and stability of a 3-guild ecological network when confronted with interaction switches. Initially, we construct niche-based networks representing pollinator-plant-herbivore interactions[34–36], followed by the implementation of adaptive interaction rewiring within these 3-guild networks. Our investigation extends to understanding how the strengths of competition, mutualism, and antagonism impact stability, measured as resilience[37], as well as network structure, encompassing nestedness[38,39] and modularity[40].

To shed light on the emergent local structural properties resulting from interaction switches, we gauge the correlation between a plant species' degree centralities in the mutualistic and antagonistic sub-networks. Additionally, we examine the correlation between the difference in these degree centralities and the plant species' per capita energy budget (derived from mutualistic interactions minus losses from antagonistic interactions). Our findings underscore the substantial influence of the balance of interaction strengths on network resilience. Importantly, we propose that the impact of adaptive interaction rewiring on the resilience of random interaction networks is contingent upon the strength of biotic interactions.

Our analyses are conducted within 3-guild networks featuring symmetric embedded sub-networks with equal network size and connectance. When the embedded sub-networks are asymmetric, characterized by unequal network size and connectance, we broaden the complexity-stability relationship to encompass the comprehensive effect of sub-network complexity (the product of network size and connectance) on the resilience of multiplex networks.

## Results

**The structure and stability of adaptive networks.** For each initial 3-guild network, we conducted $10^5$ consecutive attempts of adaptive interaction rewiring, noting that networks reached a stationary state in species biomass, network stability, sub-networks' nestedness and modularity before $10^5$ rewiring attempts (Fig. 1; Supplementary Fig. 1). The introduction of adaptive rewiring observably improved the biomass of both animal guilds, surpassing the initial dynamic equilibrium level, as depicted in Fig. 1b. In comparison to the initial random interaction network

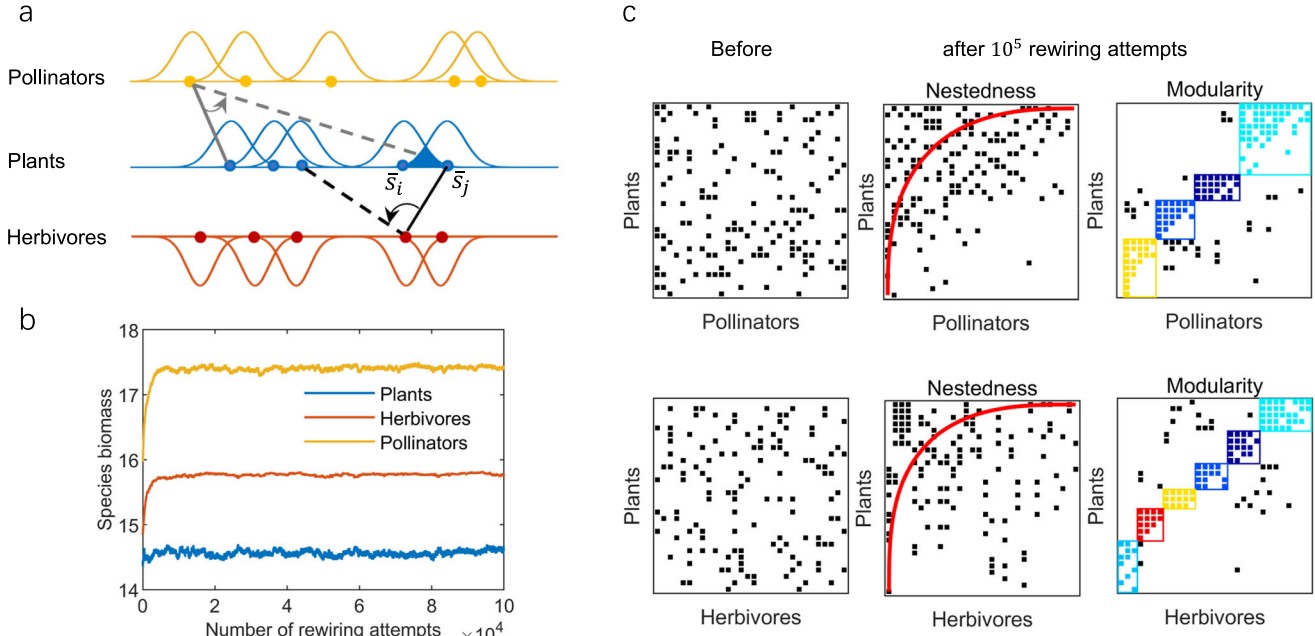

**Fig. 1 Conceptual diagram of the modeling procedure. a** Network construction from adaptive niche-based interactions. **b** Equilibrium dynamics of the total biomass of three guilds (blue: plants; orange: pollinators; red: herbivores) in networks with adaptive interaction switching, with the horizontal axis representing the number of rewiring attempts. **c** Nestedness and modularity of a plant-pollinator sub-network and a plant-herbivore sub-network before and after $10^5$ rewiring attempts for a simulation. Interaction strengths in **b** and **c** are $\{\Omega_c, \Omega_p, \Omega_m\} = \{0.1, 0.1, 0.1\}$. See Table 1 for other reference parameter values.

prior to rewiring (Fig. 1c; Supplementary Fig. 1), the mutualistic sub-network with adaptive rewiring exhibited higher levels of nestedness (z-score = 4.62) and compartmentalization (z-score = 6.36). On the other hand, the antagonistic sub-network showed increased compartmentalization (z-score = 11.49) with a non-significant change in nestedness (z-score = 0.11). Meanwhile, the minimum species biomass exhibited a positive correlation with network resilience, and species extinction probability was surprisingly low (Supplementary Fig. 2; Supplementary Table 1). Throughout the adaptive rewiring process, species richness and the total numbers of realized links in the mutualistic and antagonistic sub-networks were held constant.

The results affirm the influence of interaction strengths, denoted by the interaction composition $\{\Omega_c, \Omega_p, \Omega_m\}$, on network resilience ($-(\text{Re}(\lambda))_{\max}$), as illustrated in Fig. 2a and Supplementary Fig. 3. As anticipated, a higher strength of mutualism ($\Omega_m$) coupled with a lower strength of competition ($\Omega_c$) resulted in adaptive networks with elevated resilience (Fig. 2a). When fixing $\Omega_c$ at three levels (i.e., low, $\Omega_c = 0.01$; intermediate, $\Omega_c = 0.05$; high, $\Omega_c = 0.1$), we observed that augmenting the strength of mutualism ($\Omega_m$) stabilized the adaptive network when the competition strength was low ($\Omega_c = 0.01$), irrespective of the level of antagonism ($\Omega_p$) (Fig. 2c).

The positive association between mutualism strength and network resilience reversed to a negative correlation as the strength of competition increased to the intermediate level ($\Omega_c = 0.05$; Fig. 2d) or the high level ($\Omega_c = 0.1$; Fig. 2e). Compared to the resilience of adaptive networks, the strength of interactions exerted a consistent impact on the resilience of random interaction networks (Supplementary Fig. 4b, c–e). However, under a low competition level ($\Omega_c = 0.01$), adaptive networks tended to be less resilient than their random interaction counterparts with low mutualism (Fig. 2b; Supplementary Fig. 4c). Conversely, under an intermediate level ($\Omega_c = 0.05$; Fig. 2d, Supplementary Fig. 4d) or a high level ($\Omega_c = 0.1$; Fig. 2e,

Supplementary Fig. 4e) of competition, only a high level of mutualism could persistently maintain the negative relative resilience of adaptive networks (i.e., the resilience of the adaptive network minus the resilience of the random interaction network).

**Degree centrality of plants**. Networks resulting from adaptive interaction rewiring exhibited a notable positive correlation between a plant's degree centrality within the mutualistic guild ($d_{mut}$) and its degree centrality within the antagonistic guild ($d_{ant}$) ($\rho = 0.38, r^2 = 0.26$; Fig. 3a). In contrast, this correlation in random interaction networks with arbitrary partnerships was negligible ($\rho = 0.02, r^2 = 0.0004$; Supplementary Fig. 5a). The combinations of interaction strengths $\{\Omega_c, \Omega_p, \Omega_m\}$ in the adaptive network strongly influenced this degree centrality correlation (Fig. 3b, c). Specifically, increasing the strength of mutualism ($\Omega_m$) resulted in a higher correlation between $d_{mut}$ and $d_{ant}$, reaching a plateau (Fig. 3b), irrespective of the strength of competition or antagonism (Fig. 3b; Supplementary Fig. 6a). Across all simulated combinations of interaction strengths, the correlation remained positive and became more pronounced for low antagonistic strength and high mutualistic strength (Fig. 3c).

The difference in a plant's degree centrality in the two sub-networks ($d_{mut} - d_{ant}$) was also found to be associated with the difference in its per capita energy budget (intake minus loss, $b_{mut} - b_{ant}$). In networks with adaptive rewiring, a robust linear relationship existed between the degree centrality difference and per capita energy budget ($\rho = 2.76, r^2 = 0.78$; Fig. 3d). This linear relationship was also present in random interaction networks with arbitrary partnership, albeit comparatively weaker ($\rho = 0.99, r^2 = 0.41$; Supplementary Fig. 5b).

Further analysis using the Spearman correlation coefficient revealed that correlations between degree centrality difference and per capita energy budget varied based on the combination of interaction strengths. Specifically, the maximum Spearman correlation coefficient was attained when the strengths of mutualism and

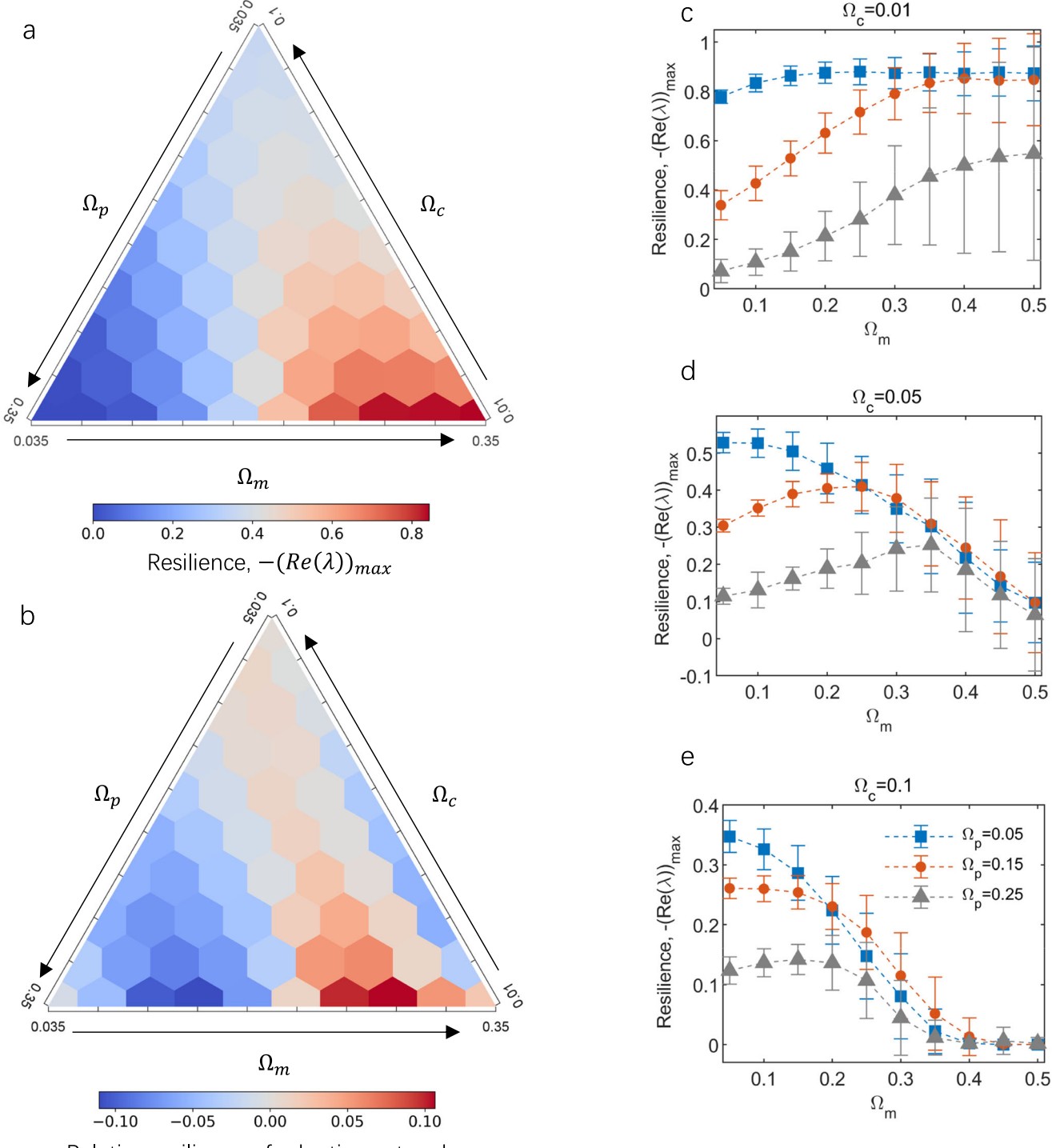

**Fig. 2 Stability response to interaction strengths.** Stability is measured as resilience, $-(Re(\lambda))_{max}$. **a** Mean resilience of adaptive networks for each combination, calculated from 60 replicates. The center of each hexagon represents an examined combination of interaction strength (55 in total); **b** Relative resilience of adaptive networks for each combination (i.e., the resilience of adaptive network minus resilience of random interaction network); **c–e** Resilience responds to increasing mutualistic strength when holding competition and antagonism constant. Curves in **c–e** show resilience with low ($\Omega_p = 0.05$; blue), intermediate ($\Omega_p = 0.15$; red) and high ($\Omega_p = 0.25$; gray) antagonistic strength. Data in **c–e** are obtained from 60 simulation replicates and presented as mean values ± SD. Other parameters are listed in Table 1.

antagonism were approximately equal, i.e., $\Omega_m \approx \Omega_p$ (Fig. 3e, f). For example, with a low strength of antagonism ($\Omega_p = 0.05$), increasing mutualistic strength led to a monotonically declining correlation, indicating a maximum correlation at low mutualistic strength (i.e., $\Omega_m = 0.05$; Fig. 3e). In contrast, when the strength of

antagonism was intermediated or high ($\Omega_p = 0.15$ or 0.25), a hump-shaped response of the correlation was observed with increasing mutualistic strength (Fig. 3e). This qualitative pattern remained robust under different levels of competition strength (Supplementary Fig. 6b). Across all simulated combinations of

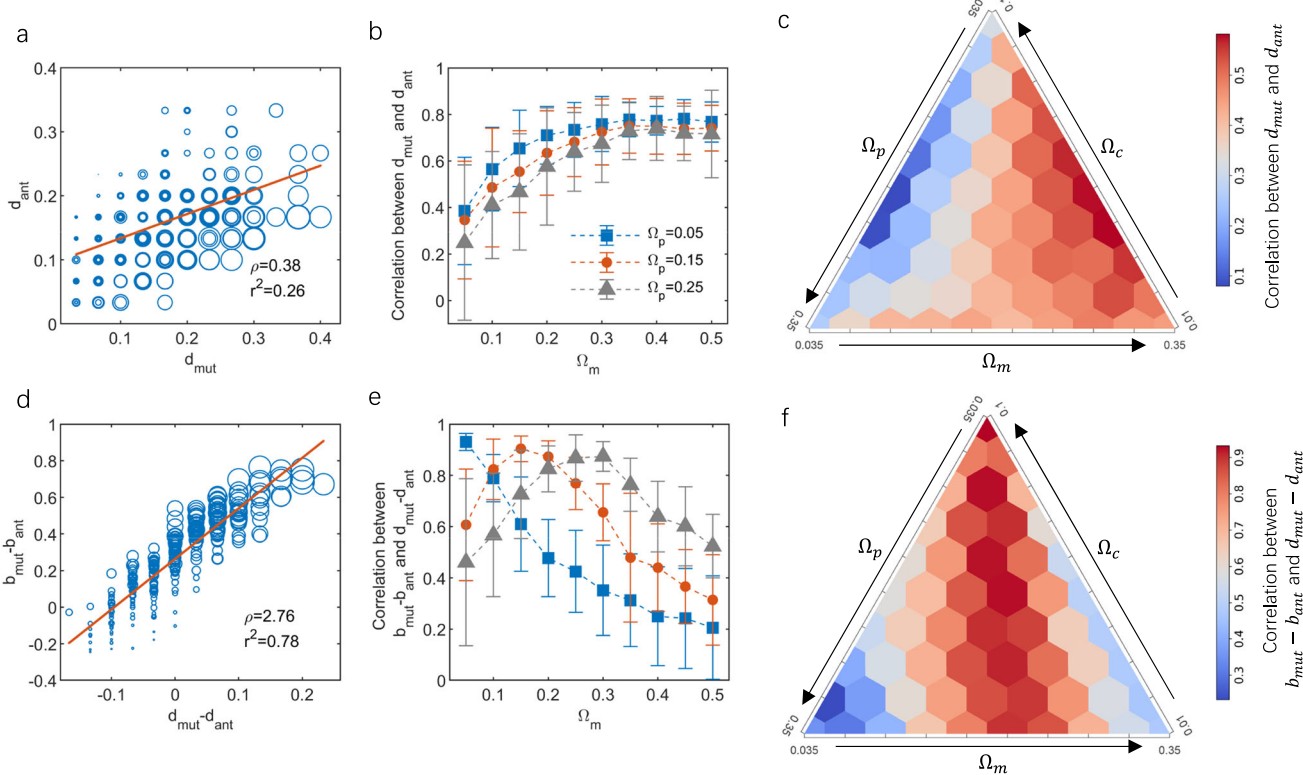

**Fig. 3 Relationships among plants' degree centralities, degree centrality difference and energy budget. a–c** Spearman correlation between a plant species' degree centralities in mutualistic and antagonistic sub-networks, (**d–f**) correlation between the plant species' degree centrality difference and its per capita energy budget, after $10^5$ rewiring attempts. Budget is calculated as energy intake from mutualism minus energy loss to antagonism. In **a** and **d**, red lines represent least-square fits and circle sizes are proportional to species' biomass ($p < 0.001$). **b, e** Effects of mutualistic strength on the mean of Spearman correlation coefficient between $d_{mut}$ and $d_{ant}$ and correlation coefficient between $b_{mut} - b_{ant}$ and $d_{mut} - d_{ant}$ (Data are obtained from 60 simulation replicates and presented as mean values ± SD). **c, f** Mean of Spearman correlation coefficients response to the combination of interaction strengths (averaged over 60 replicates). Interaction strengths in **a** and **d** are $\{\Omega_c, \Omega_p, \Omega_m\} = \{0.02, 0.175, 0.175\}$, and strength of competition in **b** and **e** is $\Omega_c = 0.1$. See Supplementary Fig. 6 for parallel results under $\Omega_c = 0.01$. Other parameters are listed in Table 1.

interaction strengths $\{\Omega_c, \Omega_p, \Omega_m\}$, the correlation between degree centrality difference and per capita energy budget for plants remained positive and reached its maximum at equal strengths of mutualistic and antagonistic interactions (Fig. 3f).

**Nestedness and modularity**. The network structure of the mutualistic and antagonistic sub-networks, as assessed by nestedness and modularity (including relative nestedness and relative modularity), was contingent on the composition of interaction strengths across the three types (Fig. 4; Supplementary Figs. 7–9). Specifically, both mutualistic and antagonistic sub-networks exhibited increased nestedness with higher mutualistic strength, irrespective of the levels of competition and antagonism strengths (Fig. 4; Supplementary Fig. 8). In contrast, in networks characterized by weak competition ($\Omega_c = 0.01$) and strong antagonism ($\Omega_p = 0.25$; Supplementary Fig. 9a, c), the modularity of both sub-networks increased with rising mutualistic strength. However, under conditions of high competition strength ($\Omega_c = 0.1$; Supplementary Fig. 9b, d), the impact of increasing mutualism on modularity was reversed.

**The role of network complexity**. These findings were consistently observed in the asymmetric 3-guild network, characterized by unequal network size and connectance within the embedded sub-networks (Supplementary Figs. 10–15). Specifically, at a low competition level ($\Omega_c = 0.01$), augmenting mutualism by increasing $\Omega_m$ contributed to stabilizing the asymmetric

3-guild network, irrespective of the complexity of the two embedded sub-networks (i.e., network resilience remained consistently higher at $\Omega_m = 0.3$ compared to $\Omega_m = 0.1$ when holding $\Omega_p$ constant; Supplementary Fig. 10a, c). However, at high competition level ($\Omega_c = 0.1$), enhancing mutualism still played a destabilizing role, aligning with observations in symmetric networks (i.e., network resilience was high at a low level of mutualism; Supplementary Fig. 10b, d).

Crucially, as the complexity in the antagonistic sub-network increased (higher $\log[S_{ant}C_{ant}]$), the resilience of 3-guild networks declined (Supplementary Fig. 10a, b). In contrast, with an increase in mutualistic complexity (higher $\log[S_{mut}C_{mut}]$), 3-guild networks became more resilient (Supplementary Fig. 10c, d). Therefore, increasing the complexity of mutualistic sub-networks bolstered the resilience of the 3-guild network, whereas increasing the complexity of antagonistic sub-networks diminished network resilience.

Increasing the complexity of an antagonistic sub-network resulted in the sub-network becoming more nested but less compartmentalized (Supplementary Fig. 11a, c). This pattern was also observed for mutualistic sub-networks (Supplementary Fig. 12a, c). However, the complexity of one sub-network did not have an impact on the structure of the other sub-network within the 3-guild network (Supplementary Figs. 13 and 14). The effects of sub-network complexity on relative nestedness ($N^*_{ant}$ and $N^*_{mut}$) and relative modularity ($Q^*_{ant}$ and $Q^*_{mut}$) were less pronounced (Supplementary Figs. 11–14), indicating that the

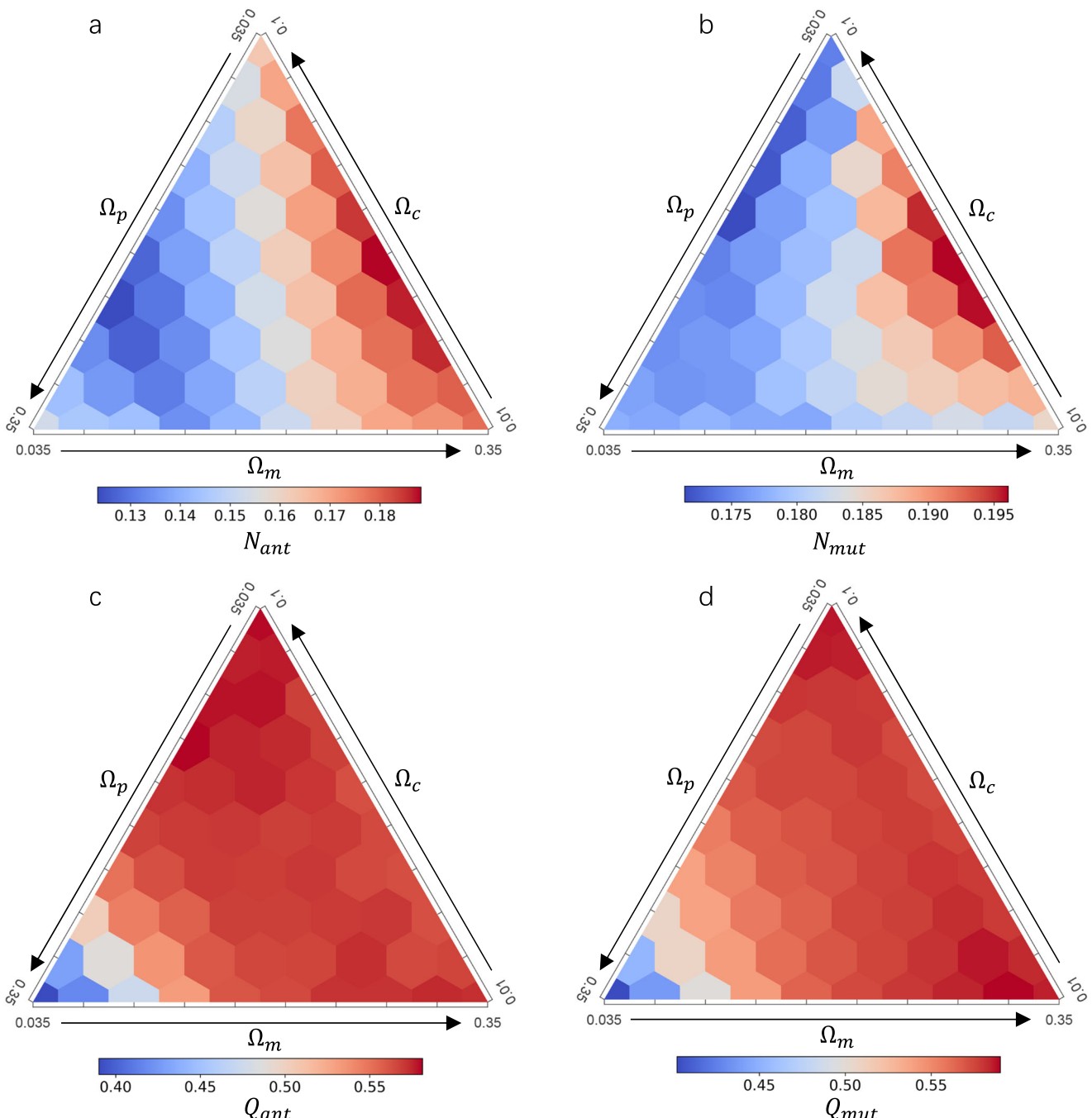

**Fig. 4 Response of nestedness and modularity to interaction strengths. a**, **b** Mean of the antagonistic nestedness ($N_{ant}$) and mutualistic nestedness ($N_{mut}$) for each combination of interaction strengths, calculated from 60 replicates. **c**, **d** Mean of the antagonistic modularity ($Q_{ant}$) and mutualistic modularity ($Q_{mut}$) for each combination of interaction strengths. See Supplementary Figs. 7–9 for the corresponding figures for relative nestedness and relative modularity, and the response of network structures to changing the mutualistic interaction strength when holding strengths of competition and antagonism constant. Other parameters are listed in Table 1.

observed effects of network complexity on network structures were primarily driven by network size and connectance.

In the asymmetric 3-guild network, both the correlation between a plant's degree centrality in the mutualistic and antagonistic sub-networks (Supplementary Fig. 15a, b) and the correlation between degree centrality difference and per capita energy budget (Supplementary Fig. 15c, d) consistently remained positive. However, these correlations did not exhibit a clear response to changes in sub-network complexity (Supplementary Fig. 15).

## Discussion

The dynamic nature of species interactions within ecological networks, such as plant-pollinator and plant-herbivore networks, is evident as a result of species turnover and interaction rewiring over time[14,29,30,41,42]. The process of interaction switching plays a pivotal role in enabling resident species within a network to finely adjust their biological niches. This, in turn, drives the emergence of network structure and has the potential to influence network stability[15–17,41].

In this study, we constructed a 3-guild network encompassing animal-plant interactions, incorporating mutualistic and antagonistic sub-networks. Our modeling approach not only considered the adaptive rewiring of species interactions but also factored in the combination of multiple interaction types. In comparison to previous literatures[2–4,15,16,28], the adaptive foraging process of interaction rewiring, coupled with the balance of interaction strengths across multiple types, clearly influenced network stability and gave rise to emergent network structures (see Figs. 2 and 3). These findings contribute valuable insights to our comprehension of the role of interaction switches in ecological networks featuring multiple interaction types, and provide a novel perspective for formulating the relationship between network structure and stability in natural communities.

The adaptive switch of interaction partners stands out as a key factor shaping ecological networks, and the ensuing local adjustment plays a crucial role in altering the structure and stability of evolutionary networks[14–18]. The ability of animal species to strategically switch interaction partners for enhanced relative benefits gives rise to a positive correlation in the generalism of plants within 3-guild networks. For instance, a mutualistic pollinator might switch to visit a more abundant or potentially more beneficial plant species. Similarly, abundant plant species are more likely to attract a diverse range of herbivory species than their rarer counterparts. This adaptive switching ability of animal species enhances relative benefits, thus contributing to the observed positive correlation in plant generalism within 3-guild networks (see Fig. 3a–c).

The manner in which plants connect with pollinators and herbivores in both mutualistic and antagonistic sub-networks aligns with empirical networks[1,4,19,31,43]. For instance, empirical studies suggest that floral traits attracting pollinators have also been shown to attract herbivores. This pattern, however, is not necessarily observed in static networks with randomly assigned interactions (see Supplementary Fig. 5). Thus, the observed pattern appears to be indicative of the key role the adaptive rewiring is at play for driving real-world network patterns. The results further suggest that the adaptive process contributes to a "the rich get richer" pattern, indicative of localized preferential attachment within pollinator-plant and herbivore-plant sub-networks[15,16,29,44]. This cumulative advantage propels adaptive networks towards greater nestedness and modularity when compared to networks with randomized interconnections[16].

Our findings illuminate a novel perspective on the correlation between the disparity in a plant's degree centrality and its per capita energy budget, revealing a discernible positive relationship. The interactions of herbivores and pollinators, while exerting conflicting impacts on the fitness of interconnecting plants[4,45], underscore a strong association between a plant's degree centrality difference and its per capita energy budget in both pollination and herbivory sub-networks (i.e., intake from pollinators and loss from herbivores).

Furthermore, the delicate balance between mutualistic and antagonistic interactions accentuates the positive correlation between a plant's degree centrality and its energy budget. In situations where plants face similar strengths of mutualism and antagonism, the theoretically expected effects of positive pollination and negative herbivory are anticipated to cancel each other out[4,45]. Consequently, the energy budget of a plant species becomes entirely contingent on its degree centralities in the two sub-networks. In essence, these results unveil a novel pathway for comprehending the non-random structure of ecological networks through the lens of interaction switches[4,14,16].

The non-random network structures resulting from adaptive interaction switches have been demonstrated to enhance network stability and ecosystem functioning[2,4,15,43,45–47]. For instance,

Suweis et al.[15] explored a mutualistic bipartite network with adaptive interaction rewiring and predicted that the adaptive network is less resilient than a random interaction network with arbitrary partnerships. In the context of a 3-guild empirical network, a positive relationship between plants' generalism regarding pollinators and herbivores was identified as a factor that could improve network stability[4]. In contrast to previous studies, our findings reveal that, despite a positive correlation between plants' dominance in pollination and herbivory sub-networks, the resilience of adaptive networks compared to random interaction networks is closely dependent on the strength of multiple interactions[16]. This dependence could be indirectly attributed to local adjustment arising from adaptive interaction rewiring[15,16].

In a less competitive network, increased mutualism may enhance species biomass, leading to a quicker recovery from perturbations in the adaptive network compared to its random counterpart. This is owing to the positive relationship between minimum species biomass and network resilience[15]. However, in a more competitive network, elevated mutualism may instigate intense competition within the same guild, driven by the "the rich get richer" pattern. This, in turn, could be detrimental to the resilience of the adaptive network.

Natural ecosystems are intricately shaped by diverse biotic interactions and the equilibrium among various functional guilds[3–9,12]. Consequently, the composition of multiple interaction strengths can disproportionately impact the resilience of adaptive networks, resulting in a nuanced influence of different interaction types on network stability. The relationship between network structure and stability in multiplex networks is intricately tied to the composition of multiple interaction types[8,12,22,23] as well as the balance of different interaction strengths[16,19,45,46].

In a previous study, an analysis of a model featuring two plant species sharing a pollinator and a herbivore underscored the pivotal role played by similar strengths of mutualism and antagonism in promoting community stability, measured as species persistence[45]. However, clear evidence supporting the idea that similar strengths of mutualistic and antagonistic interactions contribute to the enhancement of network resilience is lacking, even though this balance of strengths effectively enhances the positive correlation between a plant's degree centrality and energy budget. Consequently, our results underscore that the balance of multiple interaction types can exert selective forces that go beyond the direct additivity of different interactions[1,3,12,19,45].

Our model employed asymmetric embedded sub-networks to elucidate the robust outcomes of symmetric sub-networks, thereby underscoring the complexity-stability relationship in ecological networks involving multiple interaction types. The interplay between complexity and stability has been a long-standing focus in ecology[13,17,22,48–50]. Enhancing the complexity of pollination sub-networks has been linked to increased resilience in ecological networks, supporting a positive complexity-stability relationship[10,13,14]. Conversely, a negative relationship exists between the complexity of antagonistic sub-networks and the stability of ecological networks[3,17,49]. Kondoh[17] demonstrated that this negative complexity-stability relationship, measured as persistence, may not hold when incorporating interaction switches in antagonistic networks.

Our results suggested that the adaptive interaction switch does not alter the effects of species richness and connectance on network stability (measured as resilience) in 3-guild ecological networks, reaffirming prior research on antagonistic-mutualistic interaction networks[3]. Furthermore, some theoretical studies posit that network complexity can increase nestedness and potentially enhance the stability of mutualistic networks[10,13,14]. Our findings predicted a discernible pattern between complexity

and nestedness/modularity of mutualistic or antagonistic sub-networks. However, complexity did not effectively drive the network to become more nested or compartmentalized when compared with the null model (see Supplementary Figs. 11 and 12). These results contribute to our understanding of the weak relationships between network structure and stability in 3-guild ecological networks[3].

This study underscores the significance of niche-based inter-actions and elucidates how the adaptive rewiring of these inter-actions influences both the structures and stability of a 3-guild multiplex network. By extending existing theories on the structure-stability relationship, we unveil a more comprehensive understanding, finely tuned by the balance in the strengths of multiple interaction types. Our findings underscore that inter-action types and strengths, in conjunction with network diversity and connectivity, collectively govern the resilience of a multiplex adaptive network to perturbations.

The non-random structures revealed in our research shed light on a robust yet intricate relationship among a plant species' demographic performance, its network centrality, and its niche position[2,14,16,46]. However, the existence of a consensus on whether multiplex adaptive networks exhibit common structural properties and the nature of the relationship between these structures and network stability remains elusive[4,14–18,48]. This emerging area of multiplex networks with rewiring processes is still in its infancy, demanding further in-depth investigation. Our work lays a foundation for future explorations in this dynamic and evolving field.

## Materials and methods
**Network generation.** We examine a 3-guild ecological network comprising $S_P$ plant species, $S_M$ pollinator species, and $S_H$ herbivore species[3,9]. This network encompasses a mutualistic sub-network representing pollinator-plant interactions and an antagonistic sub-network depicting herbivore-plant interactions, with the plant guild serving as the interconnecting element between the two animal guilds[3,4,9]. Additionally, species within the same guild participate in competitive interactions[16].

Cross-guild animal-plant interactions are shaped by the niche complementarity of the interacting pair along a one-dimensional niche axis[16,34,35]. The niche profile of a species is represented by a normal distribution over the niche axis ($s$), characterized by niche breadth ($\sigma$) and central position ($\bar{s}_i$):

$$H_i(s) = \frac{1}{\sqrt{2\pi}\sigma} \exp\left(-\frac{(s - \bar{s}_i)^2}{2\sigma^2}\right) \quad (1)$$

The central position of each species is randomly generated from the interval $[0, 1]$ on the niche axis. The interspecific interaction between species $i$ and $j$ is determined by the degree of niche overlap, computed as the product of their niche profiles (Fig. 1a):

$$H_{ij} = \int H_i(s)H_j(s)ds \quad (2)$$

Consequently, the interaction strength between two species with similar positions on the niche axis is high, whereas interactions between species with distant niches on the axis are weak. Specifically, the interaction coefficient of species $j$ on $i$ is calculated as the ratio of interspecific niche overlap to conspecific niche overlap[34,35],

$$\alpha_{ij} = H_{ij}/H_{ii} = \exp\left(-\frac{(\bar{s}_i - \bar{s}_j)^2}{4\sigma^2}\right) \quad (3)$$

Expanding upon Cai et al.'s mutualistic model[16], we delineate three interaction strengths proportional to the coefficient $\alpha_{ij}$:

within-guild competitive interaction, $\beta_{ij} = \Omega_c \alpha_{ij}$ for $i \neq j$ (with $\beta_{ii} = 1$); cross-guild mutualistic interaction, $\gamma_{ij} = \Omega_m \theta_{ij} \alpha_{ij}$; and cross-guild antagonistic interaction, $\tau_{ij} = \Omega_p \theta_{ij} \alpha_{ij}$. Scalars $\Omega_c$, $\Omega_m$ and $\Omega_p$ represent the interaction strengths for perfect niche overlap and can modulate the strength of the three interspecific interactions[16,35]. The presence or absence of a cross-guild interaction ($\theta_{ij} = 1, 0$) is specified by binary matrices, $\Theta^{MP} = \{\theta_{ij}\}_{S_M \times S_P}$ for the pollinator-plant mutualistic sub-network and $\Theta^{HP} = \{\theta_{ij}\}_{S_H \times S_P}$ for the herbivore-plant antagonistic sub-network. The matrix fill of $\Theta^{MP}$ and $\Theta^{HP}$ defines the connectance for the sub-networks of cross-guild mutualistic interaction ($C_{mut}$) and antagonistic interaction ($C_{ant}$). The initial network is generated based on a random model considering arbitrary partnerships[15,16,36], where any link among animal and plant species ($\theta_{ij} = 1$) occurs with the same probability equal to the specified $C_{mut}$ or $C_{ant}$. Interaction rewiring can be implemented as matrix element reshuffling (see below).

**Community dynamics.** Building upon previous studies on eco-logical networks[13,16,18,38], we present the Lotka-Volterra model governing the dynamics of the 3-guild network:

$$\frac{dP_i}{dt} = P_i\left(r_{P_i} - \sum_{j=1}^{S_P} \beta_{ij}^P P_j + \sum_{j=1}^{S_M} \frac{\gamma_{ij} M_j}{1 + h\sum_{k \in mut(P_i)} M_k} - \sum_{j=1}^{S_H} \frac{\tau_{ij} H_j}{1 + h\sum_{k \in ant(H_j)} P_k}\right)$$
$$(4a)$$

$$\frac{dM_i}{dt} = M_i\left(r_{M_i} - \sum_{j=1}^{S_M} \beta_{ij}^M M_j + \sum_{j=1}^{S_P} \frac{\gamma_{ij} P_j}{1 + h\sum_{k \in mut(M_i)} P_k}\right) \quad (4b)$$

$$\frac{dH_i}{dt} = H_i\left(r_{H_i} - \sum_{j=1}^{S_H} \beta_{ij}^H H_j + \sum_{j=1}^{S_P} \frac{\varepsilon\tau_{ij} P_j}{1 + h\sum_{k \in ant(H_i)} P_k}\right) \quad (4c)$$

where $P_i$, $M_i$, $H_i$ $(i = 1, \ldots, S_x, x = P, M, H)$ represent the bio-mass of plant, pollinator and herbivore species $i$, respectively. Parameters $r_{P_i}$, $r_{M_i}$, $r_{H_i}$ represent the intrinsic growth rate of species $i$ in the plant, animal mutualistic, and animal antagonistic species categories. $\beta_{ij}^x$ (where $x = P, M, H$) represents intraguild competitive interaction, and $\gamma_{ij}$, $\tau_{ij}$ are defined above, with $\varepsilon$ as the conversation coefficient of predation. $mut(S_i)$ denotes the set of cross-guild mutualistic partners of species $i$, and $ant(S_i)$ denotes the set of cross-guild antagonistic partners of species $i$. It is noteworthy that mutualistic and antagonistic interactions are assumed to follow a type II functional response, shaped by the same half-saturation constant $h$ for simplicity[12,16]. Refer to Table 1 for a comprehensive list of all model parameters.

**Adaptive rewiring.** Based on previous studies implementing adaptive interaction rewiring[14–16,18], we establish the following rewiring rule. In each iteration, interaction rewiring occurs after the network settles into dynamic equilibrium. Specifically, a randomly selected animal species $i$ (either a pollinator or a her-bivore) ceases its interaction with plant species $j$ with a prob-ability of $p_{ij}$. Subsequently, it randomly explores and connects with a previously unlinked plant species $k$. The rewiring prob-ability, $p_{ij}$, is set as $1 - \psi_j^{-1}$, where $\psi_j$ represents the number of partners of plant $j$ in the same guild as species $i$. This config-uration ensures that an animal is less likely to discontinue its interaction with a more specialized plant species.

Upon connecting with a new partner, the interaction strength is updated based on niche overlap, i.e., $\gamma_{ik} = \Omega_m \alpha_{ik}$ (or $\tau_{ik} = \Omega_p \alpha_{ik}$), while simultaneously $\gamma_{ij} = 0$ (or $\tau_{ij} = 0$). The ecological network is then run using the updated quantitative

**Table 1 Parameters and their values used in the simulations.**

| Parameters | Explanations | Values in simulation |
|---|---|---|
| $r_{x_i} (x = P, M, H)$ | Per capita intrinsic growth rate of species $i$ for plants, pollinators, or herbivores | 1 |
| $S_x (x = P, M, H)$ | Species number of plants, pollinators, or herbivores | 30 |
| $\beta_{ij}^x (x = P, M, H)$ | Per capita competitive rate of species $j$ on species $i$ of the same guild | $\beta_{ij}$ equals to $\Omega_c \alpha_{ij}$ for interspecific competition, and 1 for intraspecific competition; $\alpha_{ij} = \exp\left(-\frac{(\bar{s}_i - \bar{s}_j)^2}{4\sigma^2}\right)$ |
| $\gamma_{ij}$ | Per capita mutualistic rate of pollinator species $j$ on plant species $i$ | $\Omega_m \theta_{ij} \alpha_{ij}$ |
| $\tau_{ij}$ | Per capita antagonistic rate of herbivore species $j$ on plant species $i$ | $\Omega_p \theta_{ij} \alpha_{ij}$ |
| $\varepsilon$ | Conversion coefficient of predation | 0.8 |
| $\Omega_k (k = c, m, p)$ | Interaction strength for perfect niche overlap | $\Omega_c \in [0.01, 0.1]$; $\Omega_m, \Omega_p \in [0.035, 0.35]$ |
| $h$ | Half-saturation constant for trophic and mutualistic interactions | 0.1 |
| $C_l (l = mut, ant)$ | Connectance of mutualistic or antagonistic sub-networks | 0.15 |
| $\sigma$ | Niche breadth | 0.1 |

interaction matrix, encompassing both the binary matrix and corresponding interaction strengths, until all population dynamics reach equilibrium (with a time interval of 50 between successive events of interaction rewiring). A rewiring event is accepted if the changed interaction leads to an increase in species $i$'s biomass at equilibrium; otherwise, the rewiring is rejected, and animal $i$ resumes its original interaction with plant $j$.

**Network analyses.** To assess the impact of adaptive interaction rewiring on emergent network structure, we measured various topological properties for ecological networks during the last $10^4$ rewiring attempts. Network stability, expressed as the resilience of ecological networks, gauges how quickly the system returns to equilibrium after a minor perturbation[3]. Resilience was computed as the absolute value of the largest real part of any eigenvalue of the Jacobian matrix[3], i.e., $|(\mathrm{Re}(\lambda))_{\max}|$ (Supplementary Note 1). It is worth noting that a system must be locally asymptotically stable (i.e., $\mathrm{Re}(\lambda) < 0$) before resilience can be calculated, making $|(\mathrm{Re}(\lambda))_{\max}|$ equivalent to $-(\mathrm{Re}(\lambda))_{\max}$.

Network structure was further quantified for $\Theta^{MP}$ and $\Theta^{HP}$ by nestedness (including the extent of nested sharing of interaction partners among species, measured based on the Overlap and Decreasing Fill[39]) and modularity[40] (representing the level of compartmentalization). These metrics were implemented using the open-source Matlab package BiMat[51] (Supplementary Note 2). To assess whether observed changes in nestedness are linked to species richness and network connectance[3,13], we computed the relative nestedness ($N^*$), indicating how nested the network is compared to the mean expected nestedness under a given null model (Supplementary Note 3). Similarly, we calculated the relative modularity ($Q^*$) to unveil how compartmentalized the network is compared to the null model (see Supplementary Note 3).

We further explored four scenarios of network complexity for mutualistic sub-networks (in terms of richness and connectance, $\{S_M, C_{mut}\}$) when holding the complexity of the antagonistic sub-network constant ($\{S_H, C_{ant}\}$). Similarly, we considered four scenarios for antagonistic sub-networks while holding the complexity of mutualistic sub-network constant[3]. These analyses aim to uncover insights into the 3-guild network with asymmetric sub-networks, specifically unequal network size and connectance within sub-networks (see Supplementary Note 4).

Besides nestedness and modularity, we examined the local structures of interconnecting plants in both the mutualistic and antagonistic sub-networks. Firstly, we considered the degree centrality of plants in mutualistic ($d_{mut}$) and antagonistic ($d_{ant}$) sub-networks, which indicates the importance of a plant node within a sub-network. This centrality is defined as the ratio of a plant node's degree with respect to an animal guild to its maximum possible degree[37] (which is $S_M$ for the pollinator guild and $S_H$ for the herbivore guild). Secondly, for plant species $i$, we measured the per capita energy intake from mutualistic interactions ($b_{mut}$), and the per capita energy loss from antagonistic interactions ($b_{ant}$):

$$b_{mut}^i = \sum_{j=1}^{S_M} \frac{\gamma_{ij} M_j}{1 + h \sum_{k \in mut(P_i)} M_k} \tag{5a}$$

$$b_{ant}^i = \sum_{j=1}^{S_H} \frac{\tau_{ij} H_j}{1 + h \sum_{k \in ant(H_j)} P_k} \tag{5b}$$

We computed Spearman correlations between a plant's degree centrality within the two sub-networks, as well as the relationships among a plant's degree centrality, per capita energy intake, per capita energy loss, and biomass. These analyses included correlations between ($d_{mut} - d_{ant}$) and ($b_{mut} - b_{ant}$), comparing networks with adaptive interaction rewiring versus random networks (i.e., initial networks with random counterparts after populations have equilibrated).

We conducted simulations of adaptive networks over a parameter space of interaction strengths by varying the interaction strengths $\{\Omega_c, \Omega_p, \Omega_m\}$. Specifically, we explored a range of competitive strength ($\Omega_c$) from 0.01 to 0.1 with a step of 0.01, while mutualistic and antagonistic strength ($\Omega_m$ and $\Omega_p$) varied from 0.035 to 0.35 with a step of 0.035. In total, we investigated 55 specific combinations of interaction strengths. To ensure a robust understanding of emergent network structures, we ran the model 60 times for each combination of interaction strengths.

Furthermore, for sensitivity analysis and to confirm the robustness of our main findings, we employed Latin hypercube sampling to explore a region of the parameter space[52]. This space, defined by ±30% of the baseline parameter values under symmetric sub-networks as used in the main text, encompasses six key parameters ($S_x$, $C_l$, $r_{x_i}$, $h$, $\varepsilon$ and $\sigma$). We randomly drew 10 samples of parameter combinations from this 6-dimensional space. Our analysis revealed that the main results were qualitatively robust across all values of species number ($S_x$), connectance ($C_l$), growth rate ($r_{x_i}$), half-saturation constant ($h$), conversation coefficient ($\varepsilon$), and niche breath ($\sigma$) (see Supplementary Note 5; Supplementary Table 2).

**Statistics and reproducibility**. We generated 1000 replicates of null models as a 'population', employing a one-tailed $z$-test approach to test statistical significance between the nestedness (modularity) of adaptive networks and that of null models through BiMat (Supplementary Note. 6). A $z$-score above 1.645 indicates that adaptive networks are significantly more nested or compartmentalized than the null model. We used the linear regression models to assess the relationship between a plant's degree centrality in mutualistic ($d_{mut}$) and antagonistic ($d_{ant}$) sub-networks, as well as the relationship between a plant's degree centrality ($d_{mut} - d_{ant}$) and its energy budget ($b_{mut} - b_{ant}$). We then examined Spearman correlations to reveal effects of interaction strengths on the relationship between a plant's degree centralities, and the degree centrality-energy budget relationship.

**Reporting summary**. Further information on research design is available in the Nature Portfolio Reporting Summary linked to this article.

## Data availability

Data sharing not applicable to this article as no datasets were generated or analyzed during the current study.

## Code availability

The simulation code for the adaptive interaction rewiring is available at https://doi.org/10.5281/zenodo.10459097[53].

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

## Acknowledgements
M.S. is supported by the National Natural Science Foundation of China (No. 32371555; 31770470), Natural Science Foundation of Anhui Province (No. 2308085MA09) and the State Key Laboratory of Integrated Management of Pest Insects and Rodents (No. IPM2104). C.H. acknowledges support from the South African Research Chair Initiative (National Research Foundation of South Africa, grant 89967).

## Author contributions
M.S., Q.M. and C.H. designed research, M.S. and Q.M. performed research; M.S. and C.H. wrote the manuscript with contributions from Q.M.

## Competing interests
The authors declare no competing interests.
