## [Peer Review File · Communications Biology]

Reviewers' comments:

Reviewer #1 (Remarks to the Author):

This paper aims to evaluate the effects of both mutualistic and antagonistic rewiring interactions in a pollinator-plant-herbivore tritrophic network (which is rarely done). Overall, the Authors find that mutualistic and antagonistic interactions are more influential on network stability than network configuration (nestedness/modularity) – I believe. I believe this paper is novel and interesting to a general audience and is worthy of publication – although I must admit I do not typically read the literature on ODE derived networks, but I did find it quite interesting. If anything, I hope that my comments are able to improve the manuscript.

Major comments

Are the bipartite “subnetworks” really bipartite networks - since there can be interactions between species from the same set (i.e. competition interactions) rather than only between species of different sets (as is normally the definition of bipartite networks)? Moreover, do the bipartite network metrics consider interactions between species from the same set? E.g. In Figure one (the matrix representations of modularity and nestedness): wouldn't competition interactions be ignored in any metrics here? Same applies for the connectance measure.

It appears to me that the metrics of nestedness and modularity are binary (e.g., NODF). However, there are weights to the interactions, as per your Omega and alpha values. Could you analyze your networks using weighted network metrics and would this change your result (e.g., wNODF)? Is it fair to compare the binary network metrics (i.e. nestedness/modularity) to the weighted interaction strengths and derive the conclusion that antagonistic/mutualistic weighted interactions are more important than nestedness/modularity - since you find that the interactions are significant to stability but not nestedness/modularity, could this be the reason why?

When analyzing the ODEs: can species go extinct? It appears that none do (since connectance remains the same). So I assume some species can have really low biomass values as $t \rightarrow \infty$. However, does it make sense to compare nestedness/modularity values when the interactions between species can be extremely low (i.e. changes in biomass at extremely low values), since most empirical studies that do investigate these metrics, typically rely on a discrete species interactions rather than infinitesimal small interactions? As well, do differences in initial conditions also alter your results?

Why analyze the network using subnetworks to begin with, since, as you argue, including many both mutualistic and antagonistic interactions are essential. In other words, isn't your network really a food web/multitrophic network as analyzed using stability?

Minor comments

To keep it self contained (as is the rest of this article), it might be useful to evaluate the integral on LN 93 H_{ij} since the reference of MacArthur and Levins is a slightly different integral (as far as I can tell).

I am under the impression that null model analyses were only performed for nestedness and modularity (i.e., N^* and Q^*) – although I could be wrong. Is there a reason this wasn't done for say, the eigenvalue analysis (stability)?

Why is it “-Real(leading eigenvalue)”? Isn't it simply the leading eigenvalue, regardless of multiplying

it by negative 1?

Starting on LN 138 you indicate that networks reached a stationary architecture after 5×10^4 rewiring attempts. Does this mean that the eigenvalues were also unaffected by rewirings or just that species reached their best configuration in terms of rewirings?

Why were specific values chosen in the analyses, e.g., $S_{\{X\}} = 30$ (as per Table S1). Are these biologically relevant?

I really like Table S1, and I think it would be great to include in the main manuscript.

On LN 40: I think it should be "particular" instead of "particularly".

Why investigate plant node centrality? A bit unclear from the introduction.

Starting on LN 133: "To consider an interaction rewiring adaptive, we compute species i 's biomass at equilibrium according to the updated eq (1) and accept the rewiring if species i 's biomass has increased; otherwise, the rewiring is rejected..." I think this means that a species' interaction only rewires if the changed interaction leads to an increase biomass at equilibrium, but I'm unsure what is meant by equilibrium?

Starting on LN 149: "We explored four cases of network complexity for mutualistic sub-networks (in terms of richness and connectance)..." but isn't this two?

On LN 165: the numerator has a negative tau, but isn't it just tau since you define $\tau_{\{-}}$ to be <0 ?

Starting on LN 285: "Our results confirmed the proposal that the effects of mutualistic and antagonistic interactions on resilience were mostly due to network complexity rather than network architecture (nestedness and modularity) (Fig. 2 and 4)". However, it's unclear to me how these figures show this.

On LN 320: "This could be due to that an animal..." I think it should read different.

Reviewer #2 (Remarks to the Author):

Review of the paper 'Adaptive interaction rewiring alters the structure and stability of a three-guild herbivore-plant-pollinator network'

In this paper the authors present a simulation-based exploration of the adaptive niche-based evolutionary model of Cai et al. (2020) but focusing on a specific type of network architecture comprising two interconnected bipartite networks. One of the bipartite networks is meant to represent plant-pollinator interactions and is mutualistic (i.e. the differential equations used to simulation the dynamics within it concern positive interactions: +/+). The other bipartite network comprises antagonistic interactions representing consumer-resource relationships between plants and herbivores (in this case the dynamical equations consider consumer-resource dynamics +/-). These two 'subnetworks' are linked by the basal level of plants, coupling the dynamics of herbivores and pollinators via the 'plant' trophic level.

By exploring the population and structural dynamics of these networks across different values of the parameters controlling the strength of mutualistic, antagonistic, and competitive interactions, the

authors come up with a series of expectations concerning structural features and network resilience across parameter levels. Their results suggest that resilience responds differently to the strength of mutualistic interactions depending on the strength of intra-guild competition, but this is in turn modulated by the strength of herbivory. Similarly, there is a complex relationship between these three parameters and different aspects of network structure such as nestedness and modularity (structural properties usually quantified when analysing ecological networks).

The contributions of these paper could be potentially interesting. However, given the current presentation of the paper it is hard to judge this. The paper is very poorly written in general. In its current form it reads more like a very first unfinished draft that should have undergone several further rounds of editing and re-writing before submission. It lacks detail in many crucial sections that try to argue its novelty and significance. Similarly, it is full of omissions and lack of specific explanations of methodological aspects. The same theme (lack of specific details) is also a recurrent theme in the discussion, which fails to comprehensively discuss the findings in the light of previous literature. I am more specific about all these points below, but in general, I was disappointed to read the paper after the expectations I had from the title.

More specific, albeit not at all exhaustive comments, since I believe the manuscript should change radically even before it is considered for review again.

The introduction needs a radical re-writing. The writing is very hard to follow. It is too vague in many places, lacking specific detail on main ideas, and also presenting the expectations and outcomes of the work in a very superficial way which is also hard to understand.

The methods section is full of fundamental omissions, including but not limited to:

- 1.- Which are the ecological networks referred to in line 138?
- 2.- There is no mention of how initial networks were created / generated.
- 3.- There is no explanation of the simulation protocol: How many replicates? For how long?
- 4.- What was the parameter space explored and why?
- 5.- When assuming that the system was stable to stop the adaptive process. How exactly was this 'stability' assessed? What do you mean by 'reached a stationary architecture'.
- 6.- Is there speciation in the model? In the original formulation of this model (Cai et al. 2020) there was, but it is not clear here whether the authors followed the same approach.
- 7.- Lines 147-148. These algorithms should be explained and their mathematical definitions specified.
- 8.- Lines 149-152 Not clear what this means.
- 9.- Lines 153-155 need further explanation as it is not clear what is meant here.
- 10.- There is no presentation of the statistical methods used to analyse the simulated data resulting from the simulations.

Additionally, the code used to conduct simulations should be accessible to be able to assess its validity and ability to produce the results presented.

Other comments:

Line 80 : 3-guild ... of three guilds (repetition).

All equations should be numbered.

Line 102. Niche overlap (instead of proximity)

Eq. 1. Not clear what is $-H_j$ and $+P_j$ in the functional responses.

Line 115. Remove 'number of individuals' as these equations can't simulate that.

Line 142. Emerged -> emergent

Results.

Line 172. What does this sentence mean?

Line 218 Network architecture of the mutualistic...

Line 227 The above results...

Lines 228-230. The sentence starts talking about competition but then the explanation refers to antagonistic (herbivorous) interactions. Please be clear.

Line 245. Not sure what is meant by asymmetric here.

Discussion.

In general the discussion lacks mention of many relevant previous work that is fundamental to understand the novelty of this work.

For example, Kefi's et al work on dynamics on multiplex ecological networks.

Pascual-Garcia et al.'s work on how intraguild competition modulates the effect of network architecture on mutualistic networks.

Suweis et al.'s work on how structural and dynamical properties emerge in mutualistic networks.

A discussion of all the above works and more should be added highlighting how the present findings go beyond all this previous knowledge.

Line 250. This sentence is too vague and is not explained after.

Line 269. What is stabilities? Were there many facets quantified?

Line 288-292. It is not clear how this conclusion is reached and it is misleading. There are many papers showing a strong and clear relationship between network architecture and stability from theoretical and empirical perspectives. If a statement like this is going to be included it needs to be much better supported.

Line 298. '... interaction partners has been proposed...'

Line 300. This is not really a 'network of networks'. Usually this term is used for situations in which each node of a network is itself another network, which is not the case here.

Line 310-313. Very hard to understand. Re-write / explain better.

Line 337. Knowledges -> knowledge.

Again, the list above is not exhaustive and I think this paper needs to be reviewed from scratch once it

has been brought up to good first submission standards.

Reviewer #3 (Remarks to the Author):

General comments

The manuscript explores how adaptive interaction rewiring can affect community stability (which is measured by a resilience index) and network structure/architecture of a three-guild herbivore-plant-pollinator network. The work is interesting and brings novelty by adding dynamics of adaptive interaction rewiring and by exploring how a plant species degree centrality in the mutualistic sub-network tends to correlate with the plant's degree centrality in the antagonistic sub-network. Authors also show that adaptive rewiring affects network structure and that degree centralities are correlated with the plant's per capita energy budget. The authors also make interesting points about the latter finding in the discussion section.

The manuscript is well written, but I believe that the text still requires further work to improve the clarity of the content. I found it a little difficult to follow through the goals of the work as presented in the introduction because of the constant interchange of terms. Figures are very nice and interesting, but unfortunately not fully understandable due to the lack of details in legends. Finally, I am not sure how accurate is the use of 'mutualism-antagonism continuum' in this study (please see more below).

Abstract

L8. 'its' instead of 'it'

Introduction

I think the introduction section would benefit from some re-writing, particularly to provide consistency on the use of terms that will help guide the reader through the complex work done here. Some examples: community/network stability/resilience, interaction intensity/strength, network structure and network architecture (those read very similar to a person not working on networks, so clear definitions/distinctions are needed). All these terms should be made very clear from the beginning. I also think the introduction would be greatly improved if authors could go a little further on the ecological aspects that drive net outcomes for plants in multi-guild networks. Another point of concern is that I am not sure the use of 'mutualism-antagonism continuum' is correctly used in this work. Please note that researchers usually refer to this continuum to incorporate both benefits and costs exerted by one species on others, often to make a point that most species are not fully mutualistic nor fully antagonistic (e.g. nectar robbers can contribute to pollination in some case or granivores can also disperse seeds). Here they explore a mutualistic sub-network and an antagonistic sub-network, and although they are interested on how rewiring can affect the plant's energy budget which is influenced by the balance of mutualistic/antagonistic links (in the three-guild network), this is not the same as what is understood by mutualism-antagonism continuum in the literature (including some of the literature cited in the manuscript). Also, because the continuum is seldom mentioned in the entire paper (only in the abstract and as key word), this could be just removed from the text.

Small comments:

Please be consistent on the use of sub-networks or subnetworks

L24. Please make clear what exactly authors refer to by 'community stability'

L20-24. This sentence is a bit too long. I suggest that author(s) should divide it in two. In L22 the sentence can start stating that the dynamics and structuring of these interactions are crucial to...

L20-22. Author(s) should also include a few key references for the sentence 'Plant-animal interactions are diverse and can be typically divided into mutualistic interactions such as those between plants and

pollinators (ref) and antagonistic interactions such as those between plants and herbivores (ref)'.
L28. Author(s) could start a new paragraph after 'modularity'?
L28. There is a need for reference and also to explain further animal performance
L31. I think author(s) could explain 'adaptive rewiring' further here
L32. Please specify what is meant by network structure
L35. I think the word 'disguise' could be changed to another one?
L38. I think authors could remove 'Indeed'
L44. What authors mean by interaction intensity? Is it the net outcome? The difference of mutualistic links and antagonistic links? How exactly is it measured?
L47-49. I could not understand this sentence, please re-write
L49-51. I would suggest a change here to remove a perceived intentionality from the animal's part in 'can learn and change its mutualistic and antagonistic plant partners to sustain or to enhance performance'. Authors could instead argue that the adaptive process in three-guild networks results from animals switching use of resources that favour performance enhancement. This process is often a response to a shifting ecological and environmental context. I would also explicitly add that one could expect that performance is likely to be positively related to a positive net outcome -> from the plant perspective, that would mean having more mutualistic than antagonistic links. It is also important to add a reference for the first part of this sentence
L60. 'functioning of ecosystems' Is it what authors are trying to tackle in this paper? If not, I would suggest adding here the importance of looking at adaptive rewiring in multiguild networks to assess impact on community stability, which is the goal of the study
L60-62. This point has already been raised along lines 34-35.
L63-64. something is missing here, please re-write
L66. Would it be the network complexity? If so, please make it clear
L66-67. It would be interesting to have in the text the relevance of each one of these network metrics for the study.
L67. It is not clear yet what are the three types of interactions. This is made clear in the methods but I think it should be clear by here already.
L73. Please fix 'budge'
L76-77. Please refer only to stability or to the resilience index here to avoid confusion.

Material and methods

This section is well presented and sufficiently clear, but some of the terms are not well defined. I would recommend working on the consistency of term use (as mentioned above) and also on explaining the relevance of each one of the metrics used in this study. Legends of the figures also require a more detailed description (please see more below).

L94. Position on the niche axis? Please be specific

Discussion

The discussion of the manuscript is interesting and does a good work on putting the results together in a clear way. I do believe, however, that some of the pieces of content included in the discussion (ecological aspects, in particular), should be introduced earlier in the text (the introduction would benefit from some more ecological contextualization of net outcomes and dynamics of multiguild systems). The last paragraph of the discussion could also improve to discuss how moving forward from their findings instead of presenting again a brief summary of the results.

L259. Needs reference

L267. Please specify what is meant by 'functions of an ecological network' here

L276. this sentence is a little too long and difficult to follow. Please re-write

L278. Effects on what? Please make it explicit

L289. Interaction instead of interactions

L288-291. I think this can be written in a much simpler way. It also sounds circular, because authors start on the 'trivial effect of network architectures on the complexity-stability relationship' but then

refer to 'its impacts on network architectures'.

L-300-303. I think this is a very important point that could be made already in the introduction.

L305. 'lead to a'

L305. between the dominance of a plant species in the mutualistic sub-network and in the antagonistic sub-network

L308. I would change 'be responsible' to 'contributing'

L309. which is likely the case in...

L336. No need to mention again '(implementing multiple interaction types and adaptive rewiring)'

L338. Repetitive

Legends

Legend of the figure 2 is not clear enough. Please make sure to specify what each color, symbol, axis etc refer to.

Legend of Figure 3 also needs better specification. Please specify to which lines you are referring in the text of the legend. The orange ones? Dashed or solid lines?

The legend of Figure 4 does not allow for understanding the figure. Please re-write and refer to each part of the figure specifically (a,b,c,d).

Reviewers' comments:

Reviewer #1

This paper aims to evaluate the effects of both mutualistic and antagonistic rewiring interactions in a pollinator-plant-herbivore tritrophic network (which is rarely done). Overall, the Authors find that mutualistic and antagonistic interactions are more influential on network stability than network configuration (nestedness/modularity) – I believe. I believe this paper is novel and interesting to a general audience and is worthy of publication – although I must admit I do not typically read the literature on ODE derived networks, but I did find it quite interesting. If anything, I hope that my comments are able to improve the manuscript.

Reply: Thank you for the positive assessment. We have revised the manuscript according to your comments and comments from the other reviewers.

Major comments

Q1: Are the bipartite “subnetworks” really bipartite networks - since there can be interactions between species from the same set (i.e. competition interactions) rather than only between species of different sets (as is normally the definition of bipartite networks)? Moreover, do the bipartite network metrics consider interactions between species from the same set? E.g. In Figure one (the matrix representations of modularity and nestedness): wouldn't competition interactions be ignored in any metrics here? Same applies for the connectance measure.

Reply: Thank you for your helpful comment. We agree that there is some confusion. First, the bipartite subnetworks are not real bipartite networks, because we assume the competition between species in the same set (i.e., competition within species in the same guild). Therefore, we have changed the bipartite networks into “mutualistic sub-network” and “antagonistic sub-network” through the entire manuscript. Second, network metrics did not consider interactions between species from the same set. In Figure 1, network structures (nestedness, modularity, mutualistic or antagonistic connectance C_{mut} , C_{ant}) were calculated based on plant and animal species in mutualistic or antagonistic sub-network. We have made a clear statement in the revised section “Materials and methods” (Lines 106-109; 131-133).

Q2: It appears to me that the metrics of nestedness and modularity are binary (e.g., NODF). However, there are weights to the interactions, as per your Omega and alpha values. Could you analyze your networks using weighted network metrics and would this change your result (e.g., wNODF)? Is it fair to compare the binary network metrics (i.e. nestedness/modularity) to the weighted interaction strengths and derive the conclusion that antagonistic/mutualistic weighted interactions are more important than nestedness/modularity - since you find that the interactions are significant to stability but not nestedness/modularity, could this be the reason why?

Reply: Thank you for your helpful comment. We have analysed wNodf and found that the effects of interaction strengths on wNodf are similar to Nodf (see the attached figures). Because the relative Nodf and modularity in the support Information are based on Nodf, so we have held Nodf in the revised manuscript. In the current results, interaction switches can increase the nestedness of mutualistic sub-network and modularity of antagonistic sub-network, whereas the strengths of multiple interaction types display a weak effect on Nodf (wNodf) and modularity of two sub-networks. The reason may be as followings: After 10^5 rewiring attempts, the nestedness and modularity have reached a static state. Meanwhile, change of interaction strengths in adaptive networks almost cannot alter the species richness and number of links (see the response of

comment Q3), thus nested and modular structures of two sub-networks are influenced slightly by interaction strengths.

Q3: When analyzing the ODEs: can species go extinct? It appears that none do (since connectance remains the same). So I assume some species can have really low biomass values as $t \rightarrow \infty$. However, does it make sense to compare nestedness/modularity values when the interactions between species can be extremely low (i.e. changes in biomass at extremely low values), since most empirical studies that do investigate these metrics, typically rely on a discrete species interactions rather than infinitesimal small interactions? As well, do differences in initial conditions also alter your results?

Reply: Thanks for your kind comment. We have drawn that the relationship between resilience and minimum of species biomass at the final 10^4 using 9 combinations of species interactions ($\Omega_c, \Omega_p, \Omega_m$). A positive resilience-minimum of species biomass relationship shows that very few species can go extinct if network resilience > 0 . Moreover, if species biomass is low at time t , adaptive rewiring may induce increase of the species biomass in the next rewiring process, in particular for animal species, success of adaptive rewiring can hold the species' biomass increasing. Meanwhile, we have shown that the number and frequency of species with very low biomass during the final 10^4 rewiring attempts (Table S1). Therefore, we did not consider the species extinction during the rewiring process (Cai et al., 2020). We have added the clarification in the revised manuscript (Lines 164-171; Appendix S1.3: Fig. S2, Table S1).

Moreover, to test the robustness of our main findings, we used a Latin hypercube sampling (LHS) to explore a region of the parameter space for sensitivity tests (Marino et al., 2008). LHS suggest that the initial parameter conditions can not alter our main results (see sensitivity analysis in the Supplementary information Appendix S1.4; Lines 217-225).

Q4: Why analyze the network using subnetworks to begin with, since, as you argue, including many both mutualistic and antagonistic interactions are essential. In other words, isn't your network really a food web/mutitrophic network as analyzed using stability?

Reply: We are sorry for this confusion. The ecological network merged the two sub-networks, including mutualistic and antagonistic interaction networks. For each network, species with different guild have mutualistic or antagonistic interactions, while each pairwise species with the same guild potentially can have competitive interactions due to niche overlap. Therefore, we analyzed the

network using the sub-networks to begin with. The methods also were used by previous studies, such as Sauve et al., 2014, Sauve et al., 2016 and Cai et al. 2020. Our networks are not really a food web.

Minor comments

Q5: To keep it self contained (as is the rest of this article), it might be useful to evaluate the integral on LN 93 H_{ij} since the reference of MacArthur and Levins is a slightly different integral (as far as I can tell).

Reply: Thank you for this assessment. MacArthur & Levins (1967) assumed the niche profile was characterized by a normal distribution, i.e., $U_1(R) = e^{-\frac{X^2}{H^2}}$, $U_2(R) = e^{-\frac{(X-D)^2}{H^2}}$, then the interaction

coefficient of two species was $\alpha = \frac{\int U_1(R)U_2(R)dR}{\int [U_1(R)]^2 dR} = \frac{\int e^{-\frac{X^2}{H^2}} e^{-\frac{(X-D)^2}{H^2}} dX}{\int \left[e^{-\frac{X^2}{H^2}} \right]^2 dX}$.

In our model, $H_{ij} = \int H_i(s)H_j(s)ds$ and the interaction coefficient is $\alpha_{ij} = H_{ij}/H_{ii}$, where $H_i(s) = \frac{1}{\sqrt{2\pi}\sigma} \exp\left(-\frac{(s-\bar{s}_i)^2}{2\sigma^2}\right)$ is a Gaussian function, then the integral on Lines 118 and 122 are consistent with theoretical derivation of MacArthur & Levins (1967) and Scheffer & van Nes (2006).

Q6: I am under the impression that null model analyses were only performed for nestedness and modularity (i.e., N^* and Q^*) – although I could be wrong. Is there a reason this wasn't done for say, the eigenvalue analysis (stability)?

Reply: We agree. According to your comment, we have added the eigenvalue analysis (stability) of the null model, and also compared the resilience of adaptive networks with random interaction networks in revised “Results” and “Discussion” (See the Appendix S1.5: Fig. S4; the revised Fig. 2; Lines 242-249; Lines 340-355).

Q7: Why is it “-Real (leading eigenvalue)”? Isn't it simply the leading eigenvalue, regardless of multiplying it by negative 1?

Reply: We measured network resilience, the capacity of the community to return to equilibrium after a small disturbance, expressed as the absolute value of the maximum real part of all eigenvalues of Jacobian matrix, i.e., $|\max(\text{Re}(\lambda))|$ (Gunderson, 2000; Sauve et al., 2014). A system needs to be locally asymptotically stable before calculating its resilience. If the network is locally asymptotically stable, then $(\text{Re}(\lambda))_{\max} < 0$ and $|\max(\text{Re}(\lambda))| = -(\text{Re}(\lambda))_{\max}$. Here, in order to express the system is stable before a small disturbance, we used $-(\text{Re}(\lambda))_{\max}$ to measured network resilience, then $-(\text{Re}(\lambda))_{\max} > 0$ can indicates that the origin is stable (Cai et al., 2020; see revised Fig. 2 and the code file “LV_PHM_run.m”). We have revised the expression for clarity (Lines 177-182; the revised Appendix S1.1).

Q8: Starting on LN 138 you indicate that networks reached a stationary architecture after 5×10^4 rewiring attempts. Does this mean that the eigenvalues were also unaffected by rewirings or just that species reached their best configuration in terms of rewirings?

Reply: Thank you for your comment. After 5×10^4 rewiring attempts, resilience, network structures (nestedness and modularity) and species biomass (Fig. 1b) will reach a dynamics equilibrium. We

have added these figures in the Supplementary information (Fig. S1) and added the relevant statement in the revised main text (Lines 164-168).

Q9: Why were specific values chosen in the analyses, e.g., $S_{\{X\}} = 30$ (as per Table S1). Are these biologically relevant?

Reply: Thank you for this comment. In this theoretical model, value of parameters and initial species richness are not some specific meanings. We have added sensitivity tests (see Appendix S1.4) using a Latin hypercube sampling (LHS) to confirm the main results are robust to values of parameters (i.e., species richness (S_x) and other parameter $C_l, r_{x_i}, h, \varepsilon, \sigma$; Lines 217-225). Moreover, we have considered effects of network complexity (including various species richness) on network stability and structure (Appendix S2).

Q10: I really like Table S1, and I think it would be great to include in the main manuscript.

Reply: Thank you for the kind comment. We have added the Table S1 in the main manuscript (see the revised Table 1).

Q11: On LN 40: I think it should be “particular” instead of “particularly”.

Reply: Thank you for your careful comment. We have revised the word (lines 39-40).

Q12: Why investigate plant node centrality? A bit unclear from the introduction.

Reply: We have added the result of 3-guild network in the section of Introduction that investigating plant node degree (Lines 53-56). In the symmetric embedded sub-networks, the importance of node degree is similar to node centrality. Thank you for your kind comment.

Q13: Starting on LN 133: “To consider an interaction rewiring adaptive, we compute species i 's biomass at equilibrium according to the updated eq (1) and accept the rewiring if species i 's biomass has increased; otherwise, the rewiring is rejected...” I think this means that a species' interaction only rewires if the changed interaction leads to an increase biomass at equilibrium, but I'm unsure what is meant by equilibrium?

Reply: We agree. An animal species' interaction only rewires if the changed interaction leads to an increase animal's biomass at equilibrium. Here, “equilibrium” means all population dynamics has reached equilibrium after interaction rewiring. We have clarified the expression and revised statement of interaction rewiring according to your comment (Lines 161-164).

Q14: Starting on LN 149: “We explored four cases of network complexity for mutualistic sub-networks (in terms of richness and connectance)...” but isn't this two?

Reply: Thanks for the comment. We considered the effects of network complexity on resilience and network structure with asymmetrical sub-networks. So, we explored four cases of mutualistic sub-networks when holding the complexity of antagonistic sub-network constant, and four cases for antagonistic sub-networks when holding the mutualistic sub-network constant (Sauve et al., 2014; Lines 191-196).

Q15: On LN 165: the numerator has a negative tau, but isn't it just tau since you define tau_{-} to be <0?

Reply: We agree. For clarity, we have re-defined $\tau_{ij} (> 0)$ represents the cross-guild antagonistic interaction, i.e., predatory interaction of herbivore H_j on plant P_i in Eqn.4a, and then conversion energy for reproduction of herbivore H_j is $\varepsilon\tau_{ij}$ in Eqn.4c (Lines 126, 139-141, 145-146). We have updated the Table 1.

Q16: Starting on LN 285: “Our results confirmed the proposal that the effects of mutualistic and antagonistic interactions on resilience were mostly due to network complexity rather than network architecture (nestedness and modularity) (Fig. 2 and 4)”. However, it’s unclear to me how these figures show this.

Reply: Here, we originally would like to show that network complexity displays an important role in network resilience (see the revised Fig. S10). However, relationships between network structure and network resilience are trivial, because the relative nestedness and relative modularity of sub-networks respond slightly to complexity, which indicate that the ecological network cannot become more nested or modular than null network as increasing of network complexity (Fig. S11-S14).

Now, we have re-written the section of “Discussion” and mainly discussed the emergent structures in the adaptive networks according to other reviewers’ comments. We also have re-written a discussion on the complexity-stability relationship. In particular, we have explained the trivial relationship between network structure and stability in the 3-guild ecological network (Lines 421-427).

Q17: On LN 320: “This could be due to that an animal...” I think it should read different.

Reply: We have deleted the expression and have re-written the discussion referring the maximum correlation between the difference in degree centrality of plants within pollination and herbivory sub-networks and their energy budget (Lines 395-402).

Reviewer #2

The contributions of these paper could be potentially interesting. However, given the current presentation of the paper it is hard to judge this. **The paper is very poorly written in general.** In its current form it reads more like a very first unfinished draft that should have undergone several further rounds of editing and re-writing before submission. **It lacks detail in many crucial sections that try to argue its novelty and significance.** Similarly, it is full of omissions and lack of specific explanations of methodological aspects. The same theme (lack of specific details) is also a recurrent theme in the discussion, which fails to comprehensively discuss the findings in the light of previous literature. I am more specific about all these points below, but in general, I was disappointed to read the paper after the expectations I had from the title. More specific, albeit not at all exhaustive comments, since I believe the manuscript should change radically even before it is considered for review again.

Reply: Thank you for your assessment. According to your comments, we have re-written the manuscript, especially, we have added some expression or specific explanations in “Introduction”, “Materials and methods” and “Discussion”, please see the highlighted manuscript.

“Introduction”

Q1: The introduction needs a radical re-writing. The writing is very hard to follow. It is too vague in many places, lacking specific detail on main ideas, and also presenting the expectations and outcomes of the work in a very superficial way which is also hard to understand.

Reply: Thanks for your comment. We have re-written the “Introduction” (see the highlighted “Introduction” in revised manuscript).

“Materials and methods”

Q2: Which are the ecological networks referred to in line 138?

Reply: We have specified the “ecological networks” to “each initial 3-guild network” (Lines 164-165). Meanwhile, according to your comment Q3, we have added the generation of initial networks (Lines 133-135).

Q3: There is no mention of how initial networks were created/generated.

Reply: We have added the generation of initial networks in the revised “Materials and methods” (Lines 133-135).

Q4: There is no explanation of the simulation protocol: How many replicates? For how long?

Reply: Thanks for your helpful comments. In the revised “Materials and methods”, we have added the simulation details. For each combination of species interaction strength, we have 60 replicates (Lines 216-217). In each rewiring behaviour, we ran the ecological network according to updated Eqn. 4 till all population dynamics had reached equilibrium (time interval is 20 and population can reach equilibrium after interaction rewiring) (Lines 161-162). We also have added range of the interaction strengths and its step (Lines 212-215).

Q5: What was the parameter space explored and why?

Reply: Thank you for this comment. We cannot show the parameter space in the theoretical model. However, we merged the comment with other comments of Reviewer #1 (Q3 and Q9) and have added sensitivity tests using a Latin hypercube sampling (LHS) to confirm the main results are robust to values of parameters (i.e., parameter S_x , C_l , r_{x_i} , h , ε , σ). Sensitivity analysis suggested that the parameter conditions can not alter our main results (see sensitivity analysis in the Supplementary information Appendix S1.4; Lines 217-225).

Q6: When assuming that the system was stable to stop the adaptive process. How exactly was this ‘stability’ assessed? What do you mean by ‘reached a stationary architecture’.

Reply: Thank you for your comment. After 5×10^4 rewiring attempts, resilience, network structures (nestedness and modularity), and species biomass (Fig. 1b) will reach a dynamics equilibrium. We have added these figures in the Supplementary information (Fig. S1) and added the relevant statement in the revised main text (Lines 164-168). Here, “reached a stationary architecture” means the network structure (including nestedness and modularity) reached a stable state. Please also see comment Q8 of reviewer #1.

Q7: Is there speciation in the model? In the original formulation of this model (Cai et al. 2020) there was, but it is not clear here whether the authors followed the same approach.

Reply: Thank you for your helpful comment. Our model has not considered the mutation of species’ trait, then we cannot explore the speciation in the manuscript. We only discussed how the adaptive rewiring of animal species affect the network stability and network structures.

Q8: Lines 147-148. These algorithms should be explained and their mathematical definitions specified.

Reply: Agree. We have added the mathematical definitions in the Support Information (Lines 182-191; Appendix S1.2).

Q9: Lines 149-152 Not clear what this means.

Reply: Thank you for your careful comment and we have revised the expression (Lines 191-196). Here, the complexity of two sub-networks is defined as network size (species richness) and

connectivity. Moreover, we also would like to explore the 3-guild network with asymmetric sub-networks through network complexity. Therefore, we defined four cases for the structure of the antagonistic sub-network when holding the complexity of mutualistic sub-network constant; and similarly, four cases of network complexity for mutualistic sub-networks when holding the complexity of antagonistic sub-network constant. Please also see the Support Information (Appendix S2.1).

Q10: Lines 153-155 need further explanation as it is not clear what is meant here.

Reply: Agree. We have updated the expression of relative nestedness and relative modularity in the revised “2.3 Network analysis” (Lines 186-191).

Q11: There is no presentation of the statistical methods used to analyse the simulated data resulting from the simulations. Additionally, the code used to conduct simulations should be accessible to be able to assess its validity and ability to produce the results presented.

Reply: We agree. The code has been uploaded on the <https://github.com/maqi0101/Adaptive-rewiring-code>. We provided the link website in the file of Nature portfolio. Now, we have pointed the website on line in the revised manuscript (Lines 171-173). Moreover, the statistical methods used in the manuscript is common (e.g., calculate the mean and error of resilience in 60 replicates; and correlation between plants’ degree centrality), so we have not introduced in revised manuscript (but some methods have been included in the code).

Q12: Line 80 : 3-guild ... of three guilds (repetition).

Reply: Agree. We have deleted the repetition (Line 105). Thank you for your careful comment.

Q13: All equations should be numbered.

Reply: Agree. We have added the number of all equations.

Q14: Line 102. Niche overlap (instead of proximity)

Reply: Agree. We have changed ‘niche proximity’ into ‘niche overlap’ (Line 127).

Q15: Eq. 1. Not clear what is -H_j and +P_j in the functional responses.

Reply: Term of $\tau_{ij,-}H_j$, $\tau_{ij,+}P_j$ in original Eqn. 1 represents $\tau_{ij,-}$ ($\tau_{ij,-} < 0$) multiplies H_j , and $\tau_{ij,+}$ ($\tau_{ij,+} > 0$) multiplies P_j . According to reviewer’s comment, we have used the sign of function explicitly in the revised Eqn. 4a and Eqn. 4c (Lines 139-141), i.e., revised the predation term into

$$- \sum_{j=1}^{S_H} \frac{\tau_{ij}H_j}{1+h \sum_{k \in \text{ant}(H_j)} P_k} \text{ (Eqn. 4a; } \tau_{ij} > 0) \text{ and } + \sum_{j=1}^{S_P} \frac{\varepsilon \tau_{ij}P_j}{1+h \sum_{k \in \text{ant}(H_j)} P_k} \text{ (Eqn. 4c).}$$

Q16: Line 115. Remove ‘number of individuals’ as these equations can’t simulate that.

Reply: Agree. We have removed the ‘number of individuals’ and only explain by ‘species biomass’ (Lines 142-143).

Q17: Line 142. Emerged -> emergent

Reply: Agree. We have replaced the word (Line 175).

“Results”

Q18: Line 172. What does this sentence mean?

Reply: Thanks for your kind comment. We have revised the expression for clarification (Lines 227-228). Here, Fig. 1b means that the biomass of animal species in the 3-guild network is strongly

affected by the adaptive rewiring, with the increase of rewiring steps, the biomass of herbivores and pollinators guilds increases rapidly and finally reaches a stable level.

Q19: Line 218 Network architecture of the mutualistic...

Reply: Agree. We have revised the expression, i.e., 'Network structure of the mutualistic and antagonistic sub-networks.....' (Line 280).

Q20: Line 227 The above results...

Reply: Agree. We have added the preposition 'the' (Line 289).

Q21: Lines 228-230. The sentence starts talking about competition but then the explanation refers to antagonistic (herbivorous) interactions. Please be clear.

Reply: Agree. We have revised the expression for clarity (Lines 290-296).

Q22: Line 245. Not sure what is meant by asymmetric here.

Reply: Thanks for your constructive comment. We have showed the definition of asymmetric networks (i.e., unequal network size and connectance in the two embedded sub-networks) in the revised "2.3 Network analyses", where the effects of network complexity on resilience and structures under asymmetric networks have been also explored (Lines 99-100; 196; 289-290). In the revised manuscript, we also have re-written the 'Results' and have split the section into two parts, i.e., main findings with symmetric networks and findings with asymmetric networks (Lines 227-288; 289-315).

"Discussion"

Q23: In general the discussion lacks mention of many relevant previous work that is **fundamental to understand the novelty of this work**.

For example, Kefi's et al work on dynamics on multiplex ecological networks. Pascual-Garcia et al.'s work on how intraguild competition modulates the effect of network architecture on mutualistic networks. Suweis et al.'s work on how structural and dynamical properties emerge in mutualistic networks.

A discussion of all the above works and more should be added **highlighting how the present findings go beyond all this previous knowledge**.

Reply: Agree. Thank you for your very helpful comments. We have revised the section "Discussion", in particular, we have compared the following relevant works (e.g., Kefi et al., 2017; Pascual-García & Bastolla, 2017; Sauve et al., 2016; Suweis et al., 2013) with our findings (Lines 334-340; 364-368; 374-375; 415-417).

Q24: Line 250. This sentence is too vague and is not explained after.

Reply: Agree. In the revised "Discussion", we have deleted the sentence.

Q25: Line 269. What is stabilities? Were there many facets quantified?

Reply: Agree. We have deleted the sentence. Moreover, we have clarified the term of stability (measured by resilience) used in the "Materials and methods" (Lines 89, 177-180) and "Discussion" to compare the different results with other previous literatures (e.g., Lines 404-405, 416, 419).

Q26: Line 288-292. It is not clear how this conclusion is reached and it is misleading. There are many papers showing a strong and clear relationship between network architecture and stability from theoretical and empirical perspectives. If a statement like this is going to be included it needs to be much better supported.

Reply: Thank you for your kind help. In networks with one type of species interaction, the relationship between network structure (i.e., nestedness and modularity) and stability is obvious (Okuyama & Holland, 2008; Thébault & Fontaine, 2010). However, the effects of nestedness and modularity on network stability is not clear in the ecological networks with multiple interaction types (Sauve et al., 2014). We have added the relevant expression in the revised “Discussion” (Lines 421-427). Moreover, in the revised “Discussion”, we predominantly discussed the adaptive interaction switch combined interaction strengths on network resilience, relationship between plants’ degree centrality in two sub-networks and difference in plants’ degree centrality-energy budget correlation.

Q27: Line 298. ‘... interaction partners has been proposed...’

Reply: Agree. We have revised the expression (Line 331).

Q28: Line 300. This is not really a ‘network of networks’. Usually this term is used for situations in which each node of a network is itself another network, which is not the case here.

Reply: Agree. Thank you for your helpful comment. We have moved the sentence into “Introduction” due to the following comment of Reviewer #3 and changed “network of networks” or “network of two sub-networks” into “multiplex network” in the revised manuscript (Lines 78; 429-430).

Q29: Line 310-313. Very hard to understand. Re-write / explain better.

Reply: Agree. We have re-written the expression (Lines 374-375).

Q30: Line 337. Knowledges -> knowledge.

Reply: Agree. We have deleted the sentence according to Q40 of Reviewer #3. Thank you for your kind comments.

Reviewer #3:

General comments

The manuscript explores how adaptive interaction rewiring can affect community stability (which is measured by a resilience index) and network structure/architecture of a three-guild herbivore-plant-pollinator network. The work is interesting and brings novelty by adding dynamics of adaptive interaction rewiring and by exploring how a plant species degree centrality in the mutualistic sub-network tends to correlate with the plant’s degree centrality in the antagonistic sub-network. Authors also show that adaptive rewiring affects network structure and that degree centralities are correlated with the plant’s per capita energy budget. The authors also make interesting points about the latter finding in the discussion section.

Reply: Thank you for the positive assessment. According to your following comments, we have revised the manuscript.

Q1: The manuscript is well written, but I believe that **the text still requires further work to improve the clarity of the content**. I found it a little difficult to follow through the goals of the work as presented in the introduction because of the constant interchange of terms. Figures are very nice

and interesting, but unfortunately not fully understandable due to **the lack of details in legends**. Finally, I am not sure how accurate is the use of 'mutualism-antagonism continuum' in this study (please see more below).

Reply: We agree. We have re-written the sections of "Introduction" and "Discussion". The legends of figures have been described detailly in the revised manuscript.

Q2: Abstract

L8. 'its' instead of 'it'

Reply: We agree and have revised the expression (Line 8).

Q3: Introduction

I think the introduction section would **benefit from some re-writing**, particularly to provide consistency on the use of terms that will help guide the reader through the complex work done here. Some examples: **community/network stability/resilience, interaction intensity/strength, network structure and network architecture** (those read very similar to a person not working on networks, so clear definitions/distinctions are needed). **All these terms should be made very clear from the beginning**. I also think the introduction would be greatly improved if authors could go a little further on the **ecological aspects that drive net outcomes for plants in multi-guild networks**. Another point of concern is that I am **not sure the use of 'mutualism-antagonism continuum' is correctly used in this work**. Please note that researchers usually refer to this continuum to incorporate both benefits and costs exerted by one species on others, often to make a point that most species are not fully mutualistic nor fully antagonistic (e.g. nectar robbers can contribute to pollination in some case or granivores can also disperse seeds). Here they explore a mutualistic sub-network and an antagonistic sub-network, and although they are interested on how rewiring can affect the plant's energy budget which is influenced by the balance of mutualistic/antagonistic links (in the three-guild network), this is not the same as what is understood by mutualism-antagonism continuum in the literature (including some of the literature cited in the manuscript). Also, because the **continuum is seldom mentioned** in the entire paper (only in the abstract and as key word), this could be just removed from the text.

Reply: We agree. Thank you for your helpful comment. First, we have re-written the "Introduction" and have held the terms "network stability", "network structure", and "interaction strength" consistent through the revised manuscript (e.g., Lines 24-25; 88-91; 99-100; 177-178). Moreover, we have defined the terms at the beginning of revised section "Introduction", and "Materials and methods". For instance, term adaptive interaction rewiring (i.e., interactions are reassembled over time due to species switch their interaction partners; Lines 30-31), network stability (using resilience indicates network stability; Lines 88-91; 177-178) and network structure (modularity, nestedness, plants' degree centralities in two sub-networks, and plant's degree centrality difference-per capita energy budget correlations; Lines 183-185). Moreover, we have added some ecological aspects of net outcomes for plants in multi-guild networks (Lines 51-56). Thank you for explanation of mutualism-antagonism continuum, we have replaced continuum by "a 3-guild network" or "multiplex network" through the entire manuscript (e.g., Lines 83-84; 105).

Small comments

"Introduction"

Q4: Please be consistent on the use of sub-networks or subnetworks.

Reply: We have used sub-networks in the entire revised manuscript.

Q5: L24. Please make clear what exactly authors refer to by ‘community stability’.

Reply: Thank you for the helpful comment. First, we have held the terms “network stability” consistent through the revised manuscript. Second, we have added the definition of network stability in our model (measured by resilience; Lines 89, 177-178). Some literatures have used “network resilience” or “persistence” to measure network stability (e.g., Lines 24-25; 72). We exactly showed the measure of network stability in the revised “Discussion” (e.g., Lines 405; 415-417; 419).

Q6: L20-24. This sentence is a bit too long. I suggest that author(s) should divide it in two. In L22 the sentence can start stating that the dynamics and structuring of these interactions are crucial to...

Reply: Thank you for the helpful comment. We have revised the sentence and divided it in two according to your comment (Lines 20-24).

Q7: L20-22. Author(s) should also include a few key references for the sentence ‘Plant-animal interactions are diverse and can be typically divided into mutualistic interactions such as those between plants and pollinators (ref) and antagonistic interactions such as those between plants and herbivores (ref)’.

Reply: Agree. We have added the relevant references in the sentence, in particular, we have added some reference including plant-pollinator and plant-herbivore interactions (Line 22).

Q8: L28. Author(s) could start a new paragraph after ‘modularity’?

Reply: We have re-written the “Introduction”, and have split into two issues: one issue is multiple species interaction types (Lines 38-59) and another issue is adaptive interaction switch (Lines 60-84). The first paragraph is an overview of the two issues, and we thus have not separated the first paragraph into two paragraph. Thank you for your kind comment.

Q9: L28. There is a need for reference and also to explain further animal performance.

Reply: Agree. We have added the reference and also explained **the animal performance** (i.e., reflecting an adaptive behaviour of species for enhancing the efficiency of plant resource utilisation; Lines 28-30).

Q10: L31. I think author(s) could explain ‘adaptive rewiring’ further here.

Reply: We have deleted the sentence in the revised “Introduction”, but we have added the explanation in the first sentence of adaptive rewiring (Lines 30-31).

Q11: L32. Please specify what is meant by network structure.

Reply: We have deleted the “network structure”, but have pointed the meanings of network structure in other place of the revised “Introduction” (e.g., Lines 27-28; 53).

Q12: L35. I think the word ‘disguise’ could be changed to another one?

Reply: We agree. In the revised manuscript, we have changed “disguise” to “conceal” (Line 35).

Q13: L38. I think authors could remove ‘Indeed’.

Reply: Agree. We have removed the word “Indeed” (Line 38).

Q14: L44. What authors mean by interaction intensity? Is it the net outcome? The difference of mutualistic links and antagonistic links? How exactly is it measured?

Reply: We have clarified the meanings of interaction strength (in the revised manuscript, we have used consistently the “interaction strength”). Here, interaction strengths mean the net effect of

multiple interactions' strength, so we have added the balance of these strengths (Lines 43-45). We have re-written the expression and emphasized effects of multiple interaction types on network stability, and further we have demonstrated the previous research about the composition of mutualistic links or antagonistic links in ecological networks on network stability (Lines 47-50).

Q15: L47-49. I could not understand this sentence, please re-write

Reply: Agree. We have re-written the expression and re-arranged the sentence into the next paragraph (Lines 60-64). Please also see comment Q16.

Q16: L49-51. I would suggest a change here to remove a perceived intentionality from the animal's part in 'can learn and change its mutualistic and antagonistic plant partners to sustain or to enhance performance'. Authors could instead argue that **the adaptive process in three-guild networks results from animals switching use of resources that favour performance enhancement. This process is often a response to a shifting ecological and environmental context.** I would also explicitly add that one could expect that performance is likely to be positively related to a positive net outcome -> **from the plant perspective, that would mean having more mutualistic than antagonistic links.** It is also important to add a reference for the first part of this sentence.

Reply: Thanks for your very helpful comments. We have updated the expressions according to reviewer's comments and added the reference for the first part of this sentence (Lines 60-64).

Q17: L60. 'functioning of ecosystems' Is it what authors are trying to tackle in this paper? If not, I would suggest adding here the importance of looking at adaptive rewiring in multiguild networks to assess impact on community stability, which is the goal of the study.

Reply: We agree with reviewer's comment. We have deleted the "functioning of ecosystems" and added the goal of our study in the revised "Introduction" (e.g., Lines 77-81).

Q18: L60-62. This point has already been raised along lines 34-35.

Reply: We have deleted the point in the revised manuscript.

Q19: L63-64. something is missing here, please re-write

Reply: We have re-written the sentence (Lines 85-86).

Q20: L66. Would it be the network complexity? If so, please make it clear.

Reply: We have added the network complexity into the sentence (Lines 99-102). Moreover, we have re-written the paragraph, i.e., we firstly introduced how strength of the three interaction types affect network stability (measured as community resilience) and structures (i.e., nestedness, modularity and structural properties) with symmetry embedded sub-networks, and secondly would explore the results of adaptive networks affected by network complexity (sub-network size and connectance) with asymmetrical embedded sub-networks. Thank you for your helpful comment

Q21: L66-67. It would be interesting to have in the text the relevance of each one of these network metrics for the study.

Reply: We agree. We have added the relevance of these network metrics in the revised "2.3 Network analyses" (Lines 177-178; 183-185; 187-191). For instance, network stability, expressed as the resilience of ecological networks, is a measure of how fast the system returns to its equilibrium after a small perturbation.

Q22: L67. It is not clear yet what are the three types of interactions. This is made clear in the methods but I think it should be clear by here already.

Reply: We have added the three types of species interactions (Lines 88-89).

Q23: L73. Please fix 'budge'.

Reply: Thanks for your careful comment. It's a spelling mistake. We have changed "budge" to "budget" (Line 95).

Q24: L76-77. Please refer only to stability or to the resilience index here to avoid confusion.

Reply: Agree. We have clarified network stability is measured by 'resilience' at the start (see the revised section "Introduction" and "Materials and methods"; Lines 89, 177-178), and have fixed 'resilience' in the revised 'Results'.

"Materials and methods"

Q25: This section is well presented and sufficiently clear, but some of the terms are not well defined. I would recommend working on the consistency of term use (as mentioned above) and also on explaining the relevance of each one of the metrics used in this study. Legends of the figures also require a more detailed description (please see more below).

Reply: Thank you for this constructive suggestion. We have revised the manuscript to hold the consistency of term as mentioned above and have also explained the relevance of network metrics. Please see comments Q3, Q21. We have revised the legends of the figures (see the revised Figure legends).

Q26: L94. Position on the niche axis? Please be specific.

Reply: Thanks for the comment. Position represents site on the niche axis, we have revised the sentence in detail (Lines 119-120).

"Discussion"

Q27: The discussion of the manuscript is interesting and does a good work on putting the results together in a clear way. I do believe, however, that some of the pieces of content included in the discussion (ecological aspects, in particular), should be introduced earlier in the text (the introduction would benefit from some more ecological contextualization of net outcomes and dynamics of multi-guild systems). The last paragraph of the discussion could also improve to discuss how moving forward from their findings instead of presenting again a brief summary of the results.

Reply: Thank you for this constructive suggestion. We have revised the section of "Discussion" according to your comments. First, we have added the previous research of multi-guild networks in the revised "Introduction" (e.g., Lines 47-56, 71-76), and compared with our results in the revised "Discussion" (e.g., Lines 325-328; 334-340; 364-370). Second, we have revised the last paragraph according to the comment (Lines 437-441).

Q28: L259. Needs reference.

Reply: Agree. We have added some references (Line 325).

Q29: L267. Please specify what is meant by 'functions of an ecological network' here.

Reply: We have re-written the section of "Discussion" and have deleted the expression.

Q30: L276. this sentence is a little too long and difficult to follow. Please re-write.

Reply: We have re-written the "Discussion" and mainly focused on the effects of adaptive rewiring

combined with interaction strength. Then, we have deleted the sentence.

Q31: L278. Effects on what? Please make it explicit.

Reply: We have deleted the sentence, please see Q30.

Q32: L289. Interaction instead of interactions.

Reply: We have deleted the sentence, but we have changed 'interactions strength' to 'interaction strength' in the revised manuscript (e.g., Line 326, 354).

Q33: L288-291. I think this can be written in a much simpler way. It also sounds circular, because authors start on the 'trivial effect of network architectures on the complexity-stability relationship' but then refer to 'its impacts on network architectures'.

Reply: Agree. Because effects of nestedness and modularity on complexity-stability are trivial, then we have discussed these in a simple way (Line 421-427).

Q34: L-300-303. I think this is a very important point that could be made already in the introduction.

Reply: Agree. We have merged the sentence into "Introduction" (Lines 77-81).

Q35: L305. 'lead to a'.

Reply: We agree. We have lacked the word 'to'. We have re-written the expression (Lines 364-368).

Q36: L305. between the dominance of a plant species in the mutualistic sub-network and in the antagonistic sub-network.

Reply: Thanks for your helpful comment. We have revised the expression according to your comment (Lines 364-368).

Q37: L308. I would change 'be responsible' to 'contributing'.

Reply: Agree. Thanks for your kind help. We have replaced the expression in the revised manuscript (Lines 372-375).

Q38: L309. which is likely the case in...

Reply: Agree. We have deleted the sentence and have re-written the expression (Lines 372-375).

Q39: L336. No need to mention again '(implementing multiple interaction types and adaptive rewiring)'.

Reply: Agree. We have deleted the expression in brackets (Lines 432-434).

Q40: L338. Repetitive.

Reply: Agree. We have deleted the sentence.

"Legends"

Q41: Legend of the figure 2 is not clear enough. Please make sure to specify what each color, symbol, axis etc refer to.

Reply: Agree. We have added the relative resilience of adaptive network in the revised Fig. 2 and have re-arranged the schematic guild to read the ternary plots in Support Information (Fig. S4). Moreover, we have clarified the combinations of interaction strength of Fig. 2 in the revised section of "2. Materials and methods" (Lines 212-217) and Figure legend (Lines 584-592).

Q42: Legend of Figure 3 also needs better specification. Please specify to which lines you are referring in the text of the legend. The orange ones? Dashed or solid lines?

Reply: Agree. We have revised the legend of Fig. 3 (Lines 593-602).

Q43: The legend of Figure 4 does not allow for understanding the figure. Please re-write and refer to each part of the figure specifically (a, b, c, d).

Reply: Agree. We have revised the legend of Fig. 4 according to the comment (Lines 603-608).

References in Response:

- Cai, W., Snyder, J., Hastings, A. & D'Souza, R. M. Mutualistic networks emerging from adaptive niche-based interactions. *Nat. Commun.* 11, 5470 (2020).
- Gunderson, L. H. Ecological resilience – in theory and application. *Annu. Rev. Ecol. Evol. Syst.* 31, 425-439 (2000).
- Kéfi, S., Thébault, E., Eklöf, A., Lurgi, M., Davis, A. J., Kondoh, M. & Krumins, J. A. Toward multiplex ecological networks: accounting for multiple interaction types to understand community structure and dynamics. In Moore, J. C., de Ruiter, P. C., McCann, K. S. & Wolters, V. (Eds.) *Adaptive food webs: stability and transitions of real and model ecosystems*. Cambridge University Press, 73-87 (2017).
- MacArthur, R. & Levins, R. The limiting similarity, convergence, and divergence of coexisting species. *Am. Nat.* 101, 377-385 (1967).
- Marino, S., Hogue, I. B., Ray, C. J. & Kirschner, D. E. A methodology for performing global uncertainty and sensitivity analysis in systems biology. *J. Theor. Biol.* 254, 178-196 (2008).
- Okuyama, T. & Holland, J. N. Network structural properties mediate the stability of mutualistic communities. *Ecol. Lett.* 11, 208-216 (2008).
- Pascual-García, A. & Bastolla, U. Mutualism supports biodiversity when the direct competition is weak. *Nature Communications* 8, 1-13 (2017).
- Sauve, A. M. C., Fontaine, C. & Thébault, E. Structure-stability relationships in networks combining mutualistic and antagonistic interactions. *Oikos* 123, 378-384 (2014).
- Sauve, A., Thébault, E., Pocock, M. & Fontaine, C. How plants connect pollination and herbivory networks and their contribution to community stability. *Ecology* 97, 908-917 (2016).
- Scheffer, M. & van Nes, E. H. Self-organized similarity, the evolutionary emergence of groups of similar species. *Proc. Natl Acad. Sci. USA* 103, 6230-6235 (2006).
- Suweis, S., Simini, F., Banavar, J. R. & Maritan, A. Emergence of structural and dynamical properties of ecological mutualistic networks. *Nature* 500, 449-452 (2013).
- Thébault, E. & Fontaine, C. Stability of ecological communities and the architecture of mutualistic and trophic networks. *Science* 329, 853-856 (2010).

Reviewers' comments:

Reviewer #1 (Remarks to the Author):

I appreciate the work that the Authors have done to improve their manuscript. Overall, I believe their findings to be worthy of publication.

Major comments:

I'm still confused as to why we would want to analyze nestedness, modularity, connectedness, (bipartite measures) while ignoring other types of interactions, in this 3-guild community. Isn't one of the main strengths of the paper the idea that multitrophic networks are more realistic - and ignoring these multitrophic interactions could give incorrect views about the community? I.e., LN 20-24: "Plant-animal interactions are diverse and can be typically divided into mutualistic interactions such as those between plants and pollinators and antagonistic interactions such as those between plants and herbivores. The joint importance of these species interactions are central to our understanding of network structure, stability and functioning in ecological communities." In other words, why would you analyze your system using bipartite network metrics that consider only a single type of interaction, when you already know your community has 3 guilds with more than a single type of interaction?

In some instances, I find it hard to decipher the meaning of sections of the manuscript. For instance, the Abstract is almost entirely a list of results. I think some interpretation would be necessary here. As well, in the Discussion, some paragraphs appear to be describing multiple results, making it more difficult to understand their true meaning. For example, the paragraph between LN 331-359, starts out talking about adaptive switching: "Adaptive switch of interaction partners has been proposed as a key factor to shape ecological networks, and the local optimisation would alter the stability of evolutionary networks." Note in the preceding sentence I'm not sure what local optimisation means. Then, the paragraph continues talking about past results, and eventually about 3 guild networks. But then on LN 346, the paragraph begins again referring to particular subnetworks: "For a lowly competitive network, enhancing mutualism may benefit species biomass, and thus the adaptive network recovers more rapidly from perturbations than its random counterpart because the minimum biomass of species in ecological networks is linked positively to the network resilience (Fig. 2; Figs. S2, S4)." Note in this paragraph I'm not sure what you mean by "i.e., break spontaneously the networks with more abundant generalist species and less-abundant specialist species." Altogether, for this Discussion paragraph, I think I see what you are trying to say, but it is quite dense with different ideas. Maybe consider breaking up large paragraphs into smaller ones?

Regarding the github: there is no readme file/any way of knowing how to re-run these analyses. I highly suggest making this more interpretable.

Minor comments: These relate to the interpretation/spelling of the manuscript. There are other grammatical errors that I do not list here.

On LN 38: "Most real ecological networks harbour multiple types of biotic interactions, while many species also engage in both negative and positive interspecific interactions." I think it's the case that many species engage in positive/negative types of interactions, that defines communities to have multiple interaction types.

On LN 43: "The composition and balance of strengths of the multiple interaction types in ecological networks can profoundly influence the emerged network structure and stability, while the stabilizing mechanisms for such complex networks could differ from those for networks with a type of biotic

interactions." Do you mean a single type of biotic interaction, e.g., only plant-pollinator interactions?

On LN 47: "A theoretical study has shown that there could be an "optimal" way of mixing different interaction types that maximizes network stability, i.e., stability could reach a peak at a moderate mixture of mutualistic and antagonistic interactions. Moreover, structure of multiple biotic interactions could also affect network stability." I think these sentences say the same thing? Nore sure why both are needed.

Reviewer #2 (Remarks to the Author):

Review report for the second revision of the paper "Adaptive interaction rewiring alters the structure and stability of a three-guild herbivore-plant-pollinator network"

In this paper the authors present a simulation-based exploration of the adaptive niche-based evolutionary model of Cai et al. (2020) but focusing on a specific type of network architecture comprising two interconnected bipartite networks. One of the bipartite networks is meant to represent plant-pollinator interactions and is mutualistic (i.e. the differential equations used to simulation the dynamics within it concern positive interactions: $+/+$). The other bipartite network comprises antagonistic interactions representing consumer-resource relationships between plants and herbivores (in this case the dynamical equations consider consumer-resource dynamics $+/-$). These two 'subnetworks' are linked by the basal level of plants, coupling the dynamics of herbivores and pollinators via the 'plant' trophic level.

By exploring the population and structural dynamics of these networks across different values of the parameters controlling the strength of mutualistic, antagonistic, and competitive interactions, the authors come up with a series of expectations concerning structural features and network resilience across parameter levels. Their results suggest that resilience responds differently to the strength of mutualistic interactions depending on the strength of intra-guild competition, but this is in turn modulated by the strength of herbivory. Similarly, there is a complex relationship between these three parameters and different aspects of network structure such as nestedness and modularity (structural properties usually quantified when analysing ecological networks).

Since the previous version that I reviewed, the manuscript has improved considerably. Many of my suggestions were taken on board and I am glad that the feedback was useful for the authors.

However, before making a recommendation I think there are still some points to be addressed.

Even though the writing has improved, there are still a few parts that need tweaking as some sentences are hard to understand. See detailed comments below.

The introduction needs some work around the rationale / support for the expectations presented. At the moment this is vague. I give more specific pointers below.

The methods, although much better explained in good level of detail, still need the addition of some information and some details need to be clarified.

In particular, the number of species and network connectance used for the networks needs to be mentioned, with ideally a rationale as to why these values were chosen. Related to this is the mention to the sensitivity analysis of parameters. The results of the sensitivity analysis need to be better

presented. Fig. S3 where the authors claim this is shown, does not show this. The number of replicates from the Latin hypercube is too small and the claim that the results are 'qualitatively' similar across parameter values is not well supported.

This is very important because two of these parameters (S and C) are strongly correlated with all the measures quantified in the results (resilience, nestedness, modularity).

The code presented on the github is not commented and does not include a README, which should be thoroughly comprehensive and facilitate the usage of the code to enable dissemination and replicability.

I am strongly against accepting this paper until a clear set of instructions on how to use the code, with clear specific steps on how to produce all the figures presented in the paper is clearly detailed in a README file in the github repository, as well as a thorough documentation of what each set of instructions in the code does.

In my previous review report I highlighted the lack of explanation of the statistical methods used. The authors were dismissive of this comment in their rebuttal saying that this wasn't needed. However, this still requires attention.

In particular, there are parts in the results where the authors claim that there is a 'strong linear correlation' between different measures they provide. What methods (i.e. statistical models) were used to quantify this? Where the assumptions for the model used met? And how were they assessed? All this needs to be explained in the methods.

Explain also what correlation measures were used for the correlations that are constantly mentioned in the results.

Some further specific comments are below.

Lines 2-3. It would be good to have a reason why this is the case. Also, change 'top' to something like 'main' maybe.

Line 4. ...affects the structure and stability of a 3-guild ecological network combining mutualistic...

Lines 11-12. This sentence is missing words.

Lines 13-14. This sentences is too vague. Try to add a reason and some implications of this that make this important.

Line 21. Separate by commas the 'such as those...'

Lines 24-28. Hard to understand. Revise.

Line 31. ... interactions are re-arranged over time due to species switching their...

Lines 38 and 39 repeat the same information twice.

Revise the word emerged network to emergent network throughout.

Line 45-46. Not clear what you mean here.

Line 50. 'Moreover, THE structure...'

Lines 60-62. I do not understand this sentence.

Lines 62-63. This is too vague. Add details on what you mean by performance and enhancement and how this happens.

Line 70. Random interaction networks

Lines 81-84. This sentence is too vague. A much better explanation as to why this is the main expectation and through which mechanisms is necessary here. This is the main point related to one of my main concerns above about the lack in the rationale / support for the expectations presented. It would be nice to see a link between all the information presented so far in the introduction and this main (very important) expectation.

Line 87. The network doesn't implement this. An external algorithm does.

Line 90. Hard to understand. Re-write.

Line 91. Emergent.

Line 99 When embedded sub-networks are asymmetric...

Line 105. We consider a 3-guild ecological network with S_P plant...

Line 112. normal distribution over a one-dimensional niche...

Line 115. The niche centre of each...

Line 121-124. Therefore, the interaction strength of two interactive species with similar positions on the niche axis is high, whereas interactions between species whose niches are far apart on the niche axis are weak. Specifically, the interaction coefficient of species j on i is calculated as the ratio of interspecific...

Line 135-139. Specify the parameter values for S and C. See one of the main comments above.

Line 140. Start a new subsection here: 2.2 Community Dynamics, where the dynamic model is presented.

Lines 149-151. ...and herbivore species i . Parameters $r_{(P_i)}$, $r_{(M_i)}$, $r_{(H_i)}$ represent respectively the intrinsic growth rate of species i of plant, animal mutualistic, and animal antagonistic species. β_{ij}^x ($x=P,M,H$) represents intraguild...

Line 160. While? Or after. Is the rewiring done while the dynamics are still happening (i.e. during integration)? Or after the system has achieved equilibrium?

Line 168. Make clearer that you run the model again. And what is updated is not the equation but the specification of the interaction matrix.

Line 169. In 'time interval is 20'. Not sure what this interval means. Please explain.

Lines 173-175. This figure (Fig. 1) doesn't show this. Actually, Fig 1b suggests that this stability is

reached much earlier. Way before 1×10^4 . Also, in figure 2c the label reads 10^5 rewiring attempts.

Line 189. Network structure was further quantified for Θ^{MP} and Θ^{HP} by NODF..

Lines 194-195. we further calculated the relative nestedness (N^*), a measure of how nested the network is when compared...

Lines 220. What do you mean by 'the full spectrum'? You only explore a very small fraction of the parameter space.

Line 224. Emerged -> emergent

Lines 230-232. As mentioned above, this statement is weakly supported as too few replicates of these experiments were performed and no quantitative analysis of the results is presented. It would be good to see at least a figure showing how results change across these values. This is a key issue because at least two of the properties here (S and C) have been shown to be correlated with many of the network structural and stability properties measured here.

Fig. S3 does not show these qualitative similarities across parameters values.

Reviewer #3 (Remarks to the Author):

I reviewed a previous version of the manuscript and I think this new version has improved dramatically. It seems to me that the authors did a good job after the first round of review, so congratulations on the work. However, please note that the text still requires quite some editing. I have added some very minor comments below.

L5. merging? Something sounds wrong here

L 11-12. depend on this balance for what? Please explain further

L62-63. Sounds repetitive, similar to L28-30.

L229-231. I am curious on how 'the mutualistic sub-network with adaptive rewiring was more nested (z-score= 4.62) and more compartmentalized (z-score= 6.36)? It would also be nice to see a discussion that adds some biological meaning to this particular finding.

L409. Depends instead of 'depending'?

L419. I don't think 'detriment' is commonly used as a verb anymore

L435. Please check 'is how the way'

L438-441. Please check this sentence

Reviewers' comments:

Reviewer #1 (Remarks to the Author):

I appreciate the work that the Authors have done to improve their manuscript. Overall, I believe their findings to be worthy of publication.

Reply: Thank you for your encouraging feedback. We've incorporated your insightful comments into the manuscript, aiming to enhance its overall quality and clarity. We believe these revisions significantly contribute to the robustness and coherence of our work. We look forward to any further guidance or feedback you may provide.

Major comments

Q1: I'm still confused as to why we would want to analyze nestedness, modularity, connectedness, (bipartite measures) while ignoring other types of interactions, in this 3-guild community. Isn't one of the main strengths of the paper the idea that multitrophic networks are more realistic - and ignoring these multitrophic interactions could give incorrect views about the community? I.e., LN 20-24: "Plant-animal interactions are diverse and can be typically divided into mutualistic interactions such as those between plants and pollinators and antagonistic interactions such as those between plants and herbivores. The joint importance of these species interactions are central to our understanding of network structure, stability and functioning in ecological communities." **In other words, why would you analyze your system using bipartite network metrics that consider only a single type of interaction, when you already know your community has 3 guilds with more than a single type of interaction?**

(Fontaine et al., 2011)

(Sauve et al., 2014)

(Lurgi et al., 2016)

Reply: Thank you for your valuable insights. Initially, we adopted a merged ecological network approach, akin to interconnected plant-mutualistic animal (pollinators) and plant-antagonistic animal (herbivores) interactions, as demonstrated in previous studies (Fontaine et al., 2011; Sauve et al., 2014; Sauve et al., 2016; Kéfi et al., 2017). Our analysis involved calculating network structures—connectance, nestedness, and modularity—separately for each sub-network based on bipartite network analysis. This approach allowed us to scrutinize the nuances of individual interaction types within the larger ecological context. To clarify our focus on merged sub-networks rather than multitrophic networks, we have revised the 'Introduction' section (Lines 43-51; 61-65). This amendment aims to mitigate any potential misinterpretation regarding the scope of our study. While theoretical models incorporating multitrophic networks (food webs) exist, considering both antagonistic and mutualistic interactions simultaneously, they typically do not utilize bipartite network analyses (e.g., Mougi and Kondoh, 2012; Lurgi et al., 2016). Acknowledging the merit of

these models, we express our intention to explore the realm of multiplex networks in future research (Lines 474-477). We believe this approach will provide a more comprehensive understanding of the ecological dynamics within multitrophic communities. Your feedback has been instrumental in refining our presentation and ensuring the clarity of our research focus.

Q2: In some instances, I find it hard to decipher the meaning of sections of the manuscript. For instance, the Abstract is almost entirely a list of results. **I think some interpretation would be necessary here.** As well, **in the Discussion**, some paragraphs appear to be describing multiple results, making it more difficult to **understand their true meaning.** For example, the paragraph between LN 331-359, starts out talking about adaptive switching: "Adaptive switch of interaction partners has been proposed as a key factor to shape ecological networks, and the local optimisation would alter the stability of evolutionary networks." Note in the preceding sentence I'm not sure what local optimisation means. Then, the paragraph continues talking about past results, and eventually about 3 guild networks. But then on LN 346, the paragraph begins again referring to particular subnetworks: "For a lowly competitive network, enhancing mutualism may benefit species biomass, and thus the adaptive network recovers more rapidly from perturbations than its random counterpart because the minimum biomass of species in ecological networks is linked positively to the network resilience (Fig. 2; Figs. S2, S4)." Note in this paragraph I'm not sure what you mean by "i.e., break spontaneously the networks with more abundant generalist species and less-abundant specialist species." Altogether, for this Discussion paragraph, I think I see what you are trying to say, but it is quite dense with different ideas. Maybe consider breaking up large paragraphs into smaller ones?

Reply: Thank you for your valuable feedback. We appreciate your constructive insights, and we've implemented significant revisions to address the concerns raised. In response to your observations, we have made substantial improvements to both the "Abstract" and "Discussion" sections (please refer to the revised manuscripts for all changes). Specifically, in the "Discussion" section, we have reorganized the content for better clarity and coherence. The revised structure now follows a logical progression: first discussing non-random network structure, then delving into the relationships between this structure and network stability, and finally exploring the complexity-stability dynamics concerning asymmetric sub-networks. Regarding the term "local optimisation," we have clarified its meaning as the local adjustment resulting from adaptive rewiring (Lines 371, 417). Additionally, sentences referring to past results have been either deleted or rephrased to provide a clearer explanation of our findings (Lines 419-424). The sentence involving the concept "i.e., break spontaneously the networks with more abundant generalist species and less-abundant specialist species" has been removed, and we have re-explained the pattern of "the rich get richer" for enhanced clarity (Lines 386-387). We believe these revisions significantly enhance the overall clarity and interpretability of the manuscript. Your input has been instrumental in refining our work, and we are grateful for your thoughtful engagement.

Q3: Regarding the github: there is no readme file/any way of knowing how to re-run these analyses. I highly suggest making this more interpretable.

Reply: We agree. The 'README.md' file has been uploaded, and we have added annotations in the code (<https://github.com/maqi0101/Adaptive-rewiring-model>; Lines 189-190). We also annotated the code to improve interpretability.

Draw_Ternary.py	Add files via upload	last month
LV_PHM.m	Add files via upload	last month
Model_Core_Line.m	Add files via upload	last month
Model_Core_Ternary.m	Add files via upload	last month
README.md	Update README.md	19 hours ago
cal_structure.m	Add files via upload	last month
get_jacmat.m	Add files via upload	last month
overlap.m	Add files via upload	last month

Adaptive-rewiring-model / Model_Core_Line.m

Code Blame 322 lines (301 loc) · 12.4 KB

```

0
7   Sp=30; % species number of plant
8   Sh=30; % species number of herbivore
9   Sm=30; % species number of pollinator
10  mu_r=1; % used to regulate the intrinsic growth rate of species
11  r=mu_r*ones(Sp+Sh+Sm,1); % intrinsic growth rate of species
12  h=0.1; % Half-saturation constant for trophic and mutualistic interactions
13  e=0.8; % conversation coefficient of predation
14  %%
15  cM = 0.15; % connectance of mutualistic or antagonistic subnetworks
16  trSpan = 1; % length of niche axis [0,1]
17  nw=0.1; % niche width
18  int_comp=0.01; % strength of competition
19  % int_mut=0.1; % strength of mutualism
20  % int_ant=0.1; % strength of antagonism
21  eta=1; % the parameters used to adjust the probability of disconnection
22  %%
23  int_m=0.05:0.05:0.5; % range of mutualistic strength
24  int_a=[0.05,0.15,0.25]; % range of antagonistic strength
25  %%
26  rep=60; % repetitions
27  ant_N=zeros(length(int_a),length(int_m),rep); % antagonistic nestedness
28  mut_N=zeros(length(int_a),length(int_m),rep); % mutualistic nestedness
29  ant_Q=zeros(length(int_a),length(int_m),rep); % antagonistic modularity
30  mut_Q=zeros(length(int_a),length(int_m),rep); % mutualistic modularity

```

Minor comments:

These relate to the interpretation/spelling of the manuscript. There are other grammatical errors that I do not list here.

Q4: On LN 38: "Most real ecological networks harbour multiple types of biotic interactions, while many species also engage in both negative and positive interspecific interactions." I think it's the case that many species engage in positive/negative types of interactions, that defines communities to have multiple interaction types.

Reply: Thank you for your valuable feedback. We appreciate your insights, and based on your comment, we have made revisions in the 'Introduction' to clarify that plants usually form antagonistic interactions with herbivores and mutualistic interactions with pollinators (Lines 43-44; 61-65). This adjustment aims to enhance the precision of our description regarding the types of interactions involving plants. For methodological convenience, theoretical models often partition the

merged network into sub-networks linked by shared species, as demonstrated by previous studies (e.g., Fontaine et al., 2011). This approach helps streamline the analysis and facilitates a more focused examination of specific interaction types. Please also refer to our response to Q1 of the major comments for additional context and clarification.

Q5: On LN 43: "The composition and balance of strengths of the multiple interaction types in ecological networks can profoundly influence the emerged network structure and stability, while the stabilizing mechanisms for such complex networks could differ from those for networks with a type of biotic interactions." Do you mean a single type of biotic interaction, e.g., only plant-pollinator interactions?

Reply: Certainly. Thank you for your observation. We have addressed the concern by replacing the phrase 'a single type of biotic interaction' with 'a single mutualistic or antagonistic interaction' (Lines 63-65) to provide more clarity regarding the nature of interactions under consideration. This adjustment ensures a more precise representation of our focus on either mutualistic or antagonistic interactions within the context of plant-pollinator and plant-herbivore dynamics.

Q6: On LN 47: "A theoretical study has shown that there could be an "optimal" way of mixing different interaction types that maximizes network stability, i.e., stability could reach a peak at a moderate mixture of mutualistic and antagonistic interactions. Moreover, structure of multiple biotic interactions could also affect network stability." I think these sentences say the same thing? Nore sure why both are needed.

Reply: We agree and have merged into one sentence for improved clarity (Line 59-60).

Reviewer #2:

Remarks to the Author

Since the previous version that I reviewed, the manuscript has improved considerably. Many of my suggestions were taken on board and I am glad that the feedback was useful for the authors. However, before making a recommendation I think there are still some points to be addressed.

Reply: Thank you for your positive evaluation. We have carefully revised the manuscript in accordance with the comments you provided.

Q1: Even though the writing has improved, there are still a few parts that need tweaking as some sentences are hard to understand. See detailed comments below.

Reply: Thank you for your thorough comment. We appreciate your valuable feedback, and we have made the necessary revisions to address the specific points you highlighted. We have carefully reviewed the entire revised manuscript to ensure clarity and coherence. Please refer to the revisions for a comprehensive overview of the improvements made in response to your comments.

Q2: The introduction needs some work around the rationale / support for the expectations presented. At the moment this is vague. I give more specific pointers below.

Reply: Thank you for your insightful comment. We have incorporated the suggested expressions in accordance with your feedback.

Q3: The methods, although much better explained in good level of detail, still need the addition of some information and some details need to be clarified.

Reply: Certainly. Thank you for your comment. We acknowledge your suggestion, and in response, we have provided additional information in detail, including the relevant statistical analysis, as per your comments.

Main comments

Q4: In particular, the number of species and network connectance used for the networks needs to be mentioned, with ideally a rationale as to why these values were chosen. Related to this is the mention to the sensitivity analysis of parameters. The results of the sensitivity analysis need to be better presented. Fig. S3 where the authors claim this is shown, does not show this. The number of replicates from the Latin hypercube is too small and the claim that the results are 'qualitatively' similar across parameter values is not well supported. This is very important because two of these parameters (**S and C**) are strongly correlated with all the measures quantified in the results (resilience, nestedness, modularity).

Reply: Thank you for your constructive comment. We appreciate your feedback. In response to your concerns, we have conducted additional analyses specifically addressing the parameters S (species richness) and C (connectance). The revised Figure S3 now includes results for 16 combinations of parameters, providing a comprehensive view of the effects of mutualistic and competitive strengths on resilience across different species richness levels (S=15,30,45) and subnetwork connectance values (C=0.1,0.15,0.2). This sensitivity analysis aims to demonstrate the robustness of the results presented in Figure 2 to variations in species richness, connectance, and other model parameters. We believe these additional analyses contribute valuable insights into the stability of our findings across different parameter settings. Please refer to the updated Appendix S2.2 for more detailed information.

Q5: The code presented on the github is not commented and **does not include a README**, which should be thoroughly comprehensive and facilitate the usage of the code to enable dissemination and replicability. I am strongly against accepting this paper until a clear set of instructions on how to use the code, with clear specific steps on how to produce all the figures presented in the paper is clearly detailed in a README file in the github repository, as well as a thorough documentation of what each set of instructions in the code does.

Reply: Thank you for your helpful comment. We've made the necessary updates to address your concern. The "README.md" file has been uploaded, providing clear instructions and explanations for the code. Additionally, we have included annotations in each code file for improved clarity and ease of understanding. You can find the updated code repository at <https://github.com/maqi0101/Adaptive-rewiring-model>. For further details, please refer to the response to Q3 of Reviewer #1.

 Draw_Ternary.py	Add files via upload	last month
 LV_PHM.m	Add files via upload	last month
 Model_Core_Line.m	Add files via upload	last month
 Model_Core_Ternary.m	Add files via upload	last month
 README.md	Update README.md	19 hours ago
 cal_structure.m	Add files via upload	last month
 get_jacmat.m	Add files via upload	last month
 overlap.m	Add files via upload	last month

Q6: In my previous review report I highlighted the lack of explanation of the statistical methods used. The authors were dismissive of this comment in their rebuttal saying that this wasn't needed. However, this still requires attention. In particular, there are parts in the results where the authors claim that there is a 'strong linear correlation' between different measures they provide. What methods (i.e. **statistical models**) were used to quantify this? Where the assumptions for the model used met? And how were they assessed? All this needs to be explained in the methods. Explain also what correlation measures were used for the correlations that are constantly mentioned in the results.

Reply: Thank you for your comment. We acknowledge your suggestion and have implemented the necessary revisions. Statistical methods, particularly Spearman correlation, have been explicitly described in the main text and Support Information. We have emphasized the statistical approaches in sections '2.4 Network analyses', '3. Results', and the figure legend of Figure 3 for greater clarity and transparency (Lines 204-208; 231; 298-300; 631-638).

Specific comments

Q7: Lines 2-3. It would be good to have a reason why this is the case. Also, change 'top' to something like 'main' maybe.

Reply: We have added a reason to consider adaptive rewiring in the Abstract. And we have also changed 'top' to 'main' (Lines 5-7).

Q8: Line 4. ...affects the structure and stability of a 3-guild ecological network combining mutualistic...

Reply: We have rewritten the sentence for clarity (Lines 8-10).

Q9: Lines 11-12. This sentence is missing words.

Reply: We have rewritten the entire sentence for logic flow and clarity (Lines 14-16).

Q10: Lines 13-14. This sentence is too vague. Try to add a reason and some implications of this that make this important.

Reply: We agree and have rewritten this sentence for precision and clarity (Lines 18-20).

Q11: Line 21. Separate by commas the 'such as those...'

Reply: The sentence has been rewritten for clarity (Lines 25-26).

Q12: Lines 24-28. Hard to understand. Revise.

Reply: We agree and have rewritten this sentence for precision and clarity (Lines 28-32).

Q13: Line 31. ... interactions are re-arranged over time due to species switching their...

Reply: We have rewritten the sentence for clarity (Lines 35-36).

Q14: Lines 38 and 39 repeat the same information twice.

Reply: We agree. Considering the comment and Q1 from Reviewer #1, we have refined the expression and incorporated empirical evidence to elucidate why our focus is on studying plant-animal interactions based on bipartite networks rather than multitrophic networks (Lines 43-47).

Q15: Revise the word emerged network to emergent network throughout.

Reply: Agree. We have edited the text throughout accordingly (Lines 97, 192, 242, 365).

Q16: Line 45-46. Not clear what you mean here.

Reply: We agree and have split the expression into two sentences (Lines 57-59; 61-65).

Q17: Line 50. 'Moreover, THE structure...'

Reply: Agree. We have revised the expression and deleted the sentence.

Q18: Lines 60-62. I do not understand this sentence.

Reply: Agree and we have rephrased this sentence (Lines 66-67).

Q19: Lines 62-63. This is too vague. Add details on what you mean by performance and enhancement and how this happens.

Reply: The sentence has been stated on Line 34. We deleted this to reduce repetition.

Q20: Line 70. Random interaction networkS

Reply: Agree and have edited the text throughout (e.g., Lines 75, 85, 104).

Q21: Lines 81-84. This sentence is too vague. A much better explanation as to why this is the main expectation and through which mechanisms is necessary here. This is the main point related to one of my main concerns above about the lack in the rationale / support for the expectations presented. It would be nice to see a link between all the information presented so far in the introduction and this main (very important) expectation.

Reply: Agree. We have rewritten the sentence accordingly (Lines 86-89).

Q22: Line 87. The network doesn't implement this. An external algorithm does.

Reply: Agree. We have rephrased the expression (Lines 92-94).

Q23: Line 90. Hard to understand. Re-write.

Reply: Agree. We have rewritten this sentence (Lines 94-96; 106-108).

Q24: Line 91. Emergent.

Reply: Agree. We have edited throughout (Line 97).

Q25: Line 99 When embedded sub-networks are asymmetric...

Reply: Agree. We have revised this sentence (Lines 107-111).

Q26: Line 105. We consider a 3-guild ecological network with S_P plant...

Reply: Agree. We have revised the expression in this sentence (Line 114).

Q27: Line 112. normal distribution over a one-dimensional niche...

Reply: Agree. We have revised the expression accordingly (Line 122).

Q28: Line 115. The niche centre of each...

Reply: Agree. We have revised the expression (Line 125).

Q29: Line 121-124. Therefore, the interaction strength of two interactive species with similar positions on the niche axis is high, whereas interactions between species whose niches are far apart on the niche axis are weak. Specifically, the interaction coefficient of species j on i is calculated as the ratio of interspecific...

Reply: Thank you and we have revised these sentences accordingly for clarity (Lines 129-132).

Q30: Line 135-139. Specify the parameter values for S and C. See one of the main comments above.

Reply: We have presented the values of all model parameters in Table 1, including parameter S and C in main text (Lines 162-163). Please see Q4 of the main comment.

Q31: Line 140. Start a new subsection here: 2.2 Community Dynamics, where the dynamic model is presented.

Reply: Agree. We have added a new subsection '2.2 Community Dynamics' before presenting the dynamic model (Line 148).

Q32: Lines 149-151. ...and herbivore species i . Parameters $r_{(P_i)}$, $r_{(M_i)}$, $r_{(H_i)}$ represent respectively the intrinsic growth rate of species i of plant, animal mutualistic, and animal antagonistic species. β_{ij}^x ($x=P,M,H$) represents intraguild...

Reply: We agree and have revised the expression for clarity (Lines 155-157).

Q33: Line 160. While? Or after. Is the rewiring done while the dynamics are still happening (i.e. during integration)? Or after the system has achieved equilibrium?

Reply: Yes, 'after' is exact here (Lines 166). We have rewritten this sentence for clarity.

Q34: Line 168. Make clearer that you run the model again. And what is updated is not the equation but the specification of the interaction matrix.

Reply: We agree and have revised the expression (Line 176).

Q35: Line 169. In 'time interval is 20'. Not sure what this interval means. Please explain.

Reply: Here, 'interval' represents the time interval between successive interaction rewiring. In the revision, we have increased this to 50 to ensure that the system has reached equilibrium before the next interaction rewiring (Lines 178-179). Please also can find the value in uploaded code (on line 103 of File 'Model_Core_Line.m').

Q36: Lines 173-175. This figure (Fig. 1) doesn't show this. Actually, Fig 1b suggests that this stability is reached much earlier. Way before 1×10^4 . Also, in figure 2c the label reads 10^5 rewiring attempts.

Reply: Thank you for your comment. Fig. 1b suggests that the biomass of total plants, pollinators, and herbivores reaches equilibrium before the final rewiring attempt (10^5). Fig. S1 has revealed that the stability (Resilience) and structure of adaptive networks can achieve equilibrium before the final rewiring attempts (10^5). We acknowledge that biomass under certain parameter combinations can reach equilibrium before 10^4 (Fig. 1b), but we perform additional rewiring attempts to ensure the adaptive system thoroughly reaches equilibrium. We have revised the expression accordingly (Lines 182-185).

Q37: Line 189. Network structure was further quantified for Θ^{MP} and Θ^{HP} by NODF...

Reply: We agree and have revised the expression accordingly (Lines 200-203).

Q38: Lines 194-195. we further calculated the relative nestedness (N^*), a measure of how nested the network is when compared...

Reply: Thank you for this comment. We have revised the expression accordingly (Lines 210-211).

Q39: Lines 220. What do you mean by 'the full spectrum'? You only explore a very small fraction of the parameter space.

Reply: Thank you for this comment. Here, 'the full spectrum' represents the full range of interaction strengths of the three interaction types. We have revised the expression for clarity (Lines 237-238).

Q40: Line 224. Emerged -> emergent

Reply: Agree and corrected (Line 242).

Q41: Lines 230-232. As mentioned above, this statement is weakly supported as too few replicates of these experiments were performed and no quantitative analysis of the results is presented. It would be good to see at least a figure showing how results change across these values. This is a key issue because at least two of the properties here (S and C) have been shown to be correlated with many of the network structural and stability properties measured here. Fig. S3 does not show these qualitative similarities across parameters values.

Reply: Thanks for your comment. We appreciate your suggestion and have conducted additional analyses to further support the robustness of the results presented in Fig. 2. Building upon the insights gained from Fig. 2 and the analysis of asymmetrical sub-networks provided in the Support Information (Fig. S10), we recognize the significance of parameters S and C, along with the strengths of competitive and mutualistic interactions, in influencing network resilience. To strengthen our findings, we have compared the results of symmetrical sub-networks across 16 different parameter combinations and confirmed the robustness of our results against these key parameters (refer to the revised Fig. S3 and the updated Appendix S2.2).

Reviewer #3:

Remarks to the Author

Q1: I reviewed a previous version of the manuscript and I think this new version has improved dramatically. It seems to me that the authors did a good job after the first round of review, so congratulations on the work. However, please note that the text still requires quite some editing. I have added some very minor comments below.

Reply: Thank you for your positive assessment. We have carefully considered your comments and made revisions to enhance the manuscript based on your valuable feedback.

Q2: L5. merging? Something sounds wrong here

Reply: We agree and have rewritten the sentence (Line 10).

Q3: L 11-12. depend on this balance for what? Please explain further

Reply: We agree and have rewritten the sentence (Lines 14-16).

Q4: L62-63. Sounds repetitive, similar to L28-30.

Reply: We agree and have deleted the sentence in the revised manuscript.

Q5: L229-231. I am curious on how 'the mutualistic sub-network with adaptive rewiring was more nested (z-score= 4.62) and more compartmentalized (z-score= 6.36)'? It would also be nice to see a discussion that adds some biological meaning to this particular finding.

Reply: Agree. The ability of animal species to switch interaction partners, enhancing relative benefits, contributes to the nestedness and modularity of mutualistic networks. We have incorporated a more detailed discussion of these results in the 'Discussion' section (Lines 386-390) and included statistical analyses using the z-test in the 'Materials and Methods' section for further clarification (Lines 204-208; Appendix S1.2B).

Q6: L409. Depends instead of 'depending'?

Reply: Agree. We have corrected the entire sentence for clarity (Lines 402-403).

Q7: L419. I don't think 'detriment' is commonly used as a verb anymore

Reply: We rephrased to 'be detrimental' (Line 424).

Q8: L435. Please check 'is how the way'

Reply: We agree and have rewritten the expression (Lines 380-381).

Q9: L438-441. Please check this sentence

Reply: We have restructured the "Discussion" section in response to your comments and those from other reviewers. The revised section now follows a more coherent flow: firstly, we discuss non-random network structures; secondly, we explore the relationships between non-random structure and network stability; and finally, we delve into the complexity-stability dynamics with respect to asymmetric sub-networks. Additionally, expressions related to the "positive relationship between degree centrality in antagonistic and mutualistic sub-networks" have been re-arranged for improved clarity (Lines 370-390).

References in Response:

- Bascompte, J., Jordano, P., Melián, C. J., Olesen, J. M. The nested assembly of plant-animal mutualistic networks. *PNAS* 100, 9383-9387 (2003).
- Bronstein, J. L., Wilson, W. G. & Morris, W. F. Ecological dynamics of mutualist/antagonist communities. *Am. Nat.* 162, S24-S39 (2003).
- Cai, W., Snyder, J., Hastings, A. & D'Souza, R. M. Mutualistic networks emerging from adaptive niche-based interactions. *Nat. Commun.* 11, 5470 (2020).
- Fontaine, C., Guimarães Jr, P. R., Kéfi, S., Loeuille, N., Memmott, J., van der Putten, W. H., van Veen, F. J. F. & Thébault, E. The ecological and evolutionary implications of merging different types of networks. *Eco. Lett.* 14, 1170-1181 (2011).
- Lurgi, M., Montoya, D. & Montoya, J. M. The effects of space and diversity of interaction types on the stability of complex ecological networks. *Theor. Ecol.* 9, 3-13 (2016).
- Kéfi, S., Thébault, E., Eklöf, A., Lurgi, M., Davis, A. J., Kondoh, M. & Krumins, J. A. Toward multiplex ecological networks: accounting for multiple interaction types to understand community structure and dynamics. In Moore, J. C., de Ruiter, P. C., McCann, K. S. & Wolters, V. (Eds.) *Adaptive food webs: stability and transitions of real and model ecosystems*. Cambridge University Press, 73-87 (2017).
- Sauve, A. M. C., Fontaine, C. & Thébault, E. Structure-stability relationships in networks combining mutualistic and antagonistic interactions. *Oikos* 123, 378-384 (2014).
- Sauve, A., Thébault, E., Pockock, M. & Fontaine, C. How plants connect pollination and herbivory networks and their contribution to community stability. *Ecology* 97, 908-917 (2016).
- Zhang, F., Hui, C. & Terblanche, J. S. An interaction switch predicts the nested architecture of mutualistic networks. *Ecol. Lett.* 14, 797-803 (2011).